

# CP symmetry and symplectic modular invariance

### Gui-Jun Ding[1,2]⋆, Ferruccio Feruglio[3]† and Xiang-Gan Liu[1,2]‡

**1** Peng Huanwu Center for Fundamental Theory, Hefei, Anhui 230026, China
**2** Interdisciplinary Center for Theoretical Study and Department of Modern Physics,
University of Science and Technology of China, Hefei, Anhui 230026, China
**3** Dipartimento di Fisica e Astronomia 'G. Galilei', Università di Padova
INFN, Sezione di Padova, Via Marzolo 8, I-35131 Padua, Italy

⋆ dinggj@ustc.edu.cn, † feruglio@pd.infn.it, ‡ hepliuxg@mail.ustc.edu.cn

## Abstract

We analyze CP symmetry in symplectic modular-invariant supersymmetric theories. We show that for genus $g \geq 3$ the definition of CP is unique, while two independent possibilities are allowed when $g \leq 2$. We discuss the transformation properties of moduli, matter multiplets and modular forms in the Siegel upper half plane, as well as in invariant subspaces. We identify CP-conserving surfaces in the fundamental domain of moduli space. We make use of all these elements to build a CP and symplectic invariant model of lepton masses and mixing angles, where known data are well reproduced and observable phases are predicted in terms of a minimum number of parameters.

## Contents



## 1   Introduction

Fermion masses, mixing angles and CP violating phases are tightly linked together in the present picture of particle interactions. Yet a fundamental principle explaining their origin and allowing a more economical and basic description is still lacking. The most widely explored approach to the problem is based on flavour symmetries invoked to constrain Lagrangian parameters [1]. No exact flavour symmetry does the job, however, and realistic models should rely heavily on the properties of the symmetry breaking (SB) sector, comprising a set of scalar fields whose vacuum expectation values (VEV) are suitably oriented in flavour space. To reduce the vast arbitrariness associated with this construction, where both the flavour group and the SB sector are essentially unrestricted, we recently proposed a framework defined by a set of geometrical data [2]. Scalars responsible for SB span a moduli space, a symmetric space of the type $G/K$, $G$ being a noncompact continuous group and $K$ a maximal compact subgroup of $G$. A discrete, modular subgroup $\Gamma$ of $G$, acting on $G/K$, plays the role of flavour symmetry. In this way both the flavour symmetry group and the SB sector are closely linked and cannot be arbitrarily chosen. Modular invariant supersymmetric field theories [3,4] are a particular case of this general setting, related to the choice $G = SL(2,\mathbb{R})$, $K = SO(2)$ and $\Gamma = SL(2,\mathbb{Z})$. The moduli space $G/K$ is the upper half plane and Yukawa couplings are classical modular forms [5].

This bottom-up proposal is evidently inspired by top-down considerations from string theory. In string theory Yukawa couplings are indeed field dependent quantities, specified by the background over which the string propagates. A substantial part of this background is of geometrical origin and is described by moduli, scalar fields belonging to the moduli space, which is often a symmetric space of the type $G/K$ [6]. A wealth of theoretical activity has in fact its focus on the study of Yukawa couplings in realistic string theory compactifications [7–17] and their modular properties [18–38]. Moreover, in string theory finite modular invariance is in general only a component of a bigger *Eclectic Flavour Group*, which also involves CP and an ordinary flavour group leaving moduli invariant [39–44].

Inclusion of CP transformations is an important step in a comprehensive description of particle properties. If the theory is CP-invariant, CP violation can arise only as a consequence of the choice of the vacuum. Then, in the previously discussed class of theories, CP properties depend on the chosen point in moduli space, which simultaneously controls particle masses and mixing angles. In this context most of the observed features of the fermion spectrum might be determined mostly by the vacuum, rather than by Lagrangian parameters. Moreover, the origin of fermion masses, mixing angles and phases can be fully unified.

In the presence of a discrete symmetry like the modular one, a consistent definition of CP is not granted and requires the existence of nontrivial automorphisms of the symmetry

group [45–47]. Consistent CP transformations in modular invariant supersymmetric theories dealing with a single modulus have been discussed in refs. [39, 40]. In particular, CP transformation laws of the modulus $\tau$, of chiral matter multiplets and of modular forms have been determined [48] and several models where CP is spontaneously broken have been constructed [48–54]. Within the more general case of a multidimensional moduli space, consistent CP definitions have been examined recently in the context of symplectic modular invariant theories, where the relevant flavour group is the Siegel modular group [55, 56]. Ref. [55] discusses also CP-conserving vacua in Calabi-Yau compactifications.

In this work, we reconsider CP invariance in symplectic modular invariant theories, from a bottom-up perspective. We examine thoroughly all candidate CP definitions, arising as non-trivial automorphisms of the Siegel modular group $\Gamma = Sp(2g, \mathbb{Z})$, which coincides with $SL(2, \mathbb{Z})$ when $g = 1$. We show that for genus $g \geq 3$ there is a unique automorphism suitable to be interpreted as CP, coinciding with the one identified in refs. [55,56]. On the other hand, for genus $g \leq 2$ two possibilities are allowed [1]. Moreover, beyond the action of CP on moduli space, we analyze also the correct CP transformations properties of matter multiplets and of Siegel modular forms. We also show that, beyond the surface $\text{Re}(\tau) = 0$, there are infinite CP-invariant points on the boundary of the Siegel fundamental domain. These are the ingredients needed to build concrete multi-moduli CP-invariant models and, in the final part of our paper, we propose one such model describing the lepton sector at genus $g = 2$. By making use of a minimum number of Lagrangian parameters (five, to describe twelve observable quantities), our model reproduces all known data and predicts the CP violating lepton phases, with moduli intriguingly close to a point of enhanced symmetry in moduli space.

Our paper consists of seven Sections. In Section 2, we start by reviewing the formalism of symplectic modular-invariant supersymmetric theories. In Section 3, we provide an extensive discussion of the possible CP definitions in such theories. In Section 4 we study CP-conserving points in moduli space and in Section 5 we formulate the correct definition of CP when the theory is restricted to an invariant subspace of the entire moduli space. Our model is built and analyzed in Section 6. In a final Section we present our conclusion.

## 2   Symplectic Modular Invariance

In the class of theories under consideration here, both the flavour symmetry and the fields responsible for SB have the same origin [2]. Scalars driving SB take values in a symmetric space of the type $G/K$, $G$ being some noncompact continuous group and $K$ a maximal compact subgroup of $G$. The flavour symmetry group is a discrete, modular subgroup $\Gamma$ of $G$, acting on $G/K$. Symplectic modular invariant supersymmetric theories are based on the choice $G = Sp(2g, \mathbb{R})$, $K = Sp(2g, \mathbb{R}) \cap O(2g, \mathbb{R}) = U(g)$ and $\Gamma = Sp(2g, \mathbb{Z})$. The integer $g$ ($g = 1, 2, ...$) is called genus. The moduli space $\mathcal{H}_g = Sp(2g, \mathbb{R})/U(g)$ is the Siegel upper half plane, a natural generalization of the well-known complex upper half plane $\mathcal{H}$. The Siegel modular group $\Gamma = Sp(2g, \mathbb{Z})$ arises as the duality group in string Calabi-Yau compactifications [57–64]. Siegel modular forms are relevant in the context of string one-loop corrections [65, 66].

The elements of the symplectic group $Sp(2g, \mathbb{R})$ are $2g \times 2g$ real matrices of the type

$$\gamma = \begin{pmatrix} A & B \\ C & D \end{pmatrix}, \tag{1}$$

----

[1]For $g = 1$, this alternative had already been emphasized in ref. [50].

satisfying $\gamma^t J \gamma = J$, where $J$ is the symplectic form:

$$J = \begin{pmatrix} 0 & \mathbb{1}_g \\ -\mathbb{1}_g & 0 \end{pmatrix}. \tag{2}$$

For $g = 1$, the symplectic condition on a matrix is satisfied if and only if the determinant is one, so that we have $Sp(2,\mathbb{R}) = SL(2,\mathbb{R})$. An element $\tau$ of the moduli space $\mathcal{H}_g$ is described by a symmetric complex $g \times g$ matrix $\tau$ with positive definite imaginary part:

$$\mathcal{H}_g = \left\{ \tau \in GL(g,\mathbb{C}) \,\middle|\, \tau^t = \tau, \quad \mathrm{Im}(\tau) > 0 \right\}. \tag{3}$$

Similarly to the case $G = SL(2,\mathbb{R})$, the action of $Sp(2g,\mathbb{R})$ on $\tau$ is defined as:

$$\tau \to \gamma\tau = (A\tau + B)(C\tau + D)^{-1}. \tag{4}$$

As modular group $\Gamma$ we can choose a discrete subgroup of $Sp(2g,\mathbb{R})$. A reference choice is the Siegel modular group $\Gamma_g = Sp(2g,\mathbb{Z})$, obtained from $Sp(2g,\mathbb{R})$ by restricting the elements of the matrices $A$, $B$, $C$ and $D$ in eq. (1) to integer values. A set of generators for $\Gamma_g$ is $\{S, T_i\}$:

$$S = \begin{pmatrix} 0 & \mathbb{1}_g \\ -\mathbb{1}_g & 0 \end{pmatrix}, \quad T_i = \begin{pmatrix} \mathbb{1}_g & B_i \\ 0 & \mathbb{1}_g \end{pmatrix}, \tag{5}$$

where $\{B_i\}$ is a basis for the $g \times g$ integer symmetric matrices and $S$ coincides with the invariant symplectic form $J$ satisfying $S^2 = -\mathbb{1}_{2g}$. In particular, for $g = 1$, the Siegel modular group $Sp(2,\mathbb{Z})$ coincides with the special linear group $SL(2,\mathbb{Z})$ and the generators of $\Gamma_1$ are

$$S = \begin{pmatrix} 0 & 1 \\ -1 & 0 \end{pmatrix}, \quad T = \begin{pmatrix} 1 & 1 \\ 0 & 1 \end{pmatrix}. \tag{6}$$

For the case of $g = 2$, it is convenient to choose the generators of $\Gamma_2$ as

$$S = \begin{pmatrix} 0 & \mathbb{1}_2 \\ -\mathbb{1}_2 & 0 \end{pmatrix}, \; T_1 = \begin{pmatrix} \mathbb{1}_2 & B_1 \\ 0 & \mathbb{1}_2 \end{pmatrix}, \; T_2 = \begin{pmatrix} \mathbb{1}_2 & B_2 \\ 0 & \mathbb{1}_2 \end{pmatrix}, \; T_3 = \begin{pmatrix} \mathbb{1}_2 & B_3 \\ 0 & \mathbb{1}_2 \end{pmatrix}, \tag{7}$$

with

$$B_1 = \begin{pmatrix} 1 & 0 \\ 0 & 0 \end{pmatrix}, \quad B_2 = \begin{pmatrix} 0 & 0 \\ 0 & 1 \end{pmatrix}, \quad B_3 = \begin{pmatrix} 0 & 1 \\ 1 & 0 \end{pmatrix}. \tag{8}$$

Under $S$ and $T_i$ transformations, we find:

$$\tau \xrightarrow{S} -\tau^{-1}, \quad \tau \xrightarrow{T_i} \tau + B_i. \tag{9}$$

Other discrete subgroups of $G = Sp(2g,\mathbb{R})$, of direct interest here, are the principal congruence subgroups $\Gamma_g(n)$ of level $n$, defined as:

$$\Gamma_g(n) = \left\{ \gamma \in \Gamma_g \,\middle|\, \gamma \equiv \mathbb{1}_{2g} \,\mathrm{mod}\, n \right\}, \tag{10}$$

where $n$ is a generic positive integer, and $\Gamma_g(1) = \Gamma_g$. The group $\Gamma_g(n)$ is a normal subgroup of $\Gamma_g$, and the quotient group $\Gamma_{g,n} = \Gamma_g/\Gamma_g(n)$, which is known as finite Siegel modular group, has finite order [67,68].

## 2.1 Fundamental domain

Symplectic modular invariance can be seen as a gauge symmetry related to the redundancy of the description of physical vacua. Thus it is useful to introduce a fundamental domain $\mathcal{F}_g$ describing the set of inequivalent vacua. This is essentially the quotient between $\mathcal{H}_g$ and $\Gamma_g$. More precisely, a fundamental domain $\mathcal{F}_g$ in $\mathcal{H}_g$ for the Siegel modular group $\Gamma_g$ is a connected region of $\mathcal{H}_g$ such that each point of $\mathcal{H}_g$ can be mapped into $\mathcal{F}_g$ by a $\Gamma_g$ transformation, but no two points in the interior of $\mathcal{F}_g$ are related under $\Gamma_g$. It is considerably more complicated than the $g = 1$ case. A fundamental domain $\mathcal{F}_g$ for the action of $\Gamma_g$ on $\mathcal{H}_g$ can be defined as follows [69]:

$$
\mathcal{F}_g = \left\{ \tau \in \mathcal{H}_g \,\middle|\, \begin{cases} \text{Im}(\tau)\,\text{is reduced in the sense of Minkowski,} \\ |\det(C\tau + D)| \geq 1 \quad \text{for all } \gamma \in \Gamma_g, \\ |\text{Re}(\tau_{ij})| \leq 1/2\,; \end{cases} \right\}. \tag{11}
$$

Here Minkowski reduced means that $\text{Im}(\tau)$ satisfies the two properties:

1) $h^t \text{Im}(\tau) h \geq \text{Im}(\tau)_{kk} \quad (\forall h = (h_1, \ldots, h_g) \in \mathbb{Z}^g)$ for $1 \leq k \leq g$ whenever $h_1, \ldots, h_g$ are coprime ;

2) $\text{Im}(\tau)_{k,k+1} \geq 0$ for $0 \leq k \leq g-1$.

In his book [69] Siegel proved that for any genus $g$ such a fundamental domain is determined by only finitely many inequalities of the form $|\det(C\tau + D)| \geq 1$ and with the Minkowski condition. The boundary $\partial \mathcal{F}_g$ is defined as the set of points in $\mathcal{F}_g$, where at least one of the relations in eq. (11) is realized as an equality. In general points lying on the boundary $\partial \mathcal{F}_g$ are related by Siegel modular transformations. The boundary of the fundamental domain $\mathcal{F}_g$ for $g > 1$ is very complex. At genus $g = 2$ we parametrize the moduli $\tau$ as

$$
\tau = \begin{pmatrix} \tau_1 & \tau_3 \\ \tau_3 & \tau_2 \end{pmatrix}. \tag{12}
$$

The fundamental domain $\mathcal{F}_2$ can be defined by the following inequalities [70,71]:

$$
\mathcal{F}_2 = \left\{ \tau \in \mathcal{H}_2 \,\middle|\, \begin{cases} |\text{Re}(\tau_1)| \leq 1/2, \quad |\text{Re}(\tau_3)| \leq 1/2. \quad |\text{Re}(\tau_2)| \leq 1/2, \\ \text{Im}(\tau_2) \geq \text{Im}(\tau_1) \geq 2\text{Im}(\tau_3) \geq 0, \\ |\tau_1| \geq 1, \quad |\tau_2| \geq 1, \quad |\tau_1 + \tau_2 - 2\tau_3 \pm 1| \geq 1, \\ |\det(\tau + \mathcal{E}_i)| \geq 1, \end{cases} \right\}, \tag{13}
$$

where the set $\{\mathcal{E}_i\}$ includes the following 15 matrices:

$$
\begin{pmatrix} 0 & 0 \\ 0 & 0 \end{pmatrix}, \quad \begin{pmatrix} \pm 1 & 0 \\ 0 & 0 \end{pmatrix}, \quad \begin{pmatrix} 0 & 0 \\ 0 & \pm 1 \end{pmatrix}, \quad \begin{pmatrix} \pm 1 & 0 \\ 0 & \pm 1 \end{pmatrix},
$$
$$
\begin{pmatrix} \pm 1 & 0 \\ 0 & \mp 1 \end{pmatrix}, \quad \begin{pmatrix} 0 & \pm 1 \\ \pm 1 & 0 \end{pmatrix}, \quad \begin{pmatrix} \pm 1 & \pm 1 \\ \pm 1 & 0 \end{pmatrix}, \quad \begin{pmatrix} 0 & \pm 1 \\ \pm 1 & \pm 1 \end{pmatrix}. \tag{14}
$$

When one of these inequality is satisfied as an equality, we recover a polynomial equation in $\text{Re}(\tau_i)$ and $\text{Im}(\tau_i)$, defining a real 5-dimensional wall. From eq. (13) we count 28 walls, that determine the boundary $\partial \mathcal{F}_2$ of the domain $\mathcal{F}_2$.

## 2.2 Siegel modular forms

Another important building block of the theory are the Siegel modular forms, holomorphic complex functions of the variables $\tau$, enjoying good transformation properties under the Siegel modular group $\Gamma_g$. They are specified by the genus $g$, the weight $k$ and the level $n$, $k$ and $n$ being non-negative integers. Siegel modular forms $Y(\tau)$ of integral weight $k$ and level $n$ at genus $g$ are holomorphic functions on the Siegel upper half plane $\mathcal{H}_g$ transforming under $\Gamma_g(n)$ as

$$Y(\gamma\tau) = [\det(C\tau + D)]^k Y(\tau), \qquad \gamma = \begin{pmatrix} A & B \\ C & D \end{pmatrix} \in \Gamma_g(n). \tag{15}$$

When $n = 1, 2$, we have $-\mathbb{1}_{2g} \in \Gamma_g(n)$ and the above definition gives [2]:

$$Y(-\mathbb{1}_{2g}\tau) = Y(\tau) = (-1)^{kg} Y(\tau). \tag{16}$$

Therefore Siegel modular forms at genus $g$ of weight $k$ and level $n = 1, 2$ vanish if $kg$ is odd. The complex linear space $\mathcal{M}_k(\Gamma_g(n))$ of Siegel modular forms of given weight $k$, level $n$ and genus $g$ is finite dimensional and there are no non-vanishing forms of negative weight [72].

Similarly to the case $g = 1$ [5], it is possible to choose a basis $\{Y_i(\tau)\}$ in the space $\mathcal{M}_k(\Gamma_g(n))$ such that the action of $\Gamma_g$ on the elements of the basis is described by a unitary representation $\rho_{\mathbf{r}}$ of the finite Siegel modular group $\Gamma_{g,n} = \Gamma_g/\Gamma_g(n)$:

$$Y_i(\gamma\tau) = [\det(C\tau + D)]^k \rho_{\mathbf{r}}(\gamma)_{ij} Y_j(\tau), \qquad \gamma = \begin{pmatrix} A & B \\ C & D \end{pmatrix} \in \Gamma_g. \tag{17}$$

At variance with eq. (15), where only transformations of $\Gamma_g(n)$ were considered, in the previous equation the full Siegel modular group $\Gamma_g$ is acting. Eq. (17) shows that the Siegel modular forms $\{Y_i(\tau)\}$ of given weight, level and genus have good transformation properties also with respect to $\Gamma_g$.

## 2.3 Symplectic modular invariant supersymmetric theory

To build a symplectic modular invariant supersymmetric theory, we have to specify the action of $\Gamma_g$ on the matter multiplets $\varphi$ of the theory, which can belong to separate sectors $\{\varphi^{(I)}\}$. To this purpose we choose a particular level $n$. Both the genus $g$ and the level $n$ are kept fixed in the construction. The supermultiplets $\varphi^{(I)}$ of each sector $I$ are assumed to transform in a representation $\rho^{(I)}$ of the finite Siegel modular group $\Gamma_{g,n}$, with a weight $k_I$ [2].

$$\begin{cases} \tau \to \gamma\tau = (A\tau + B)(C\tau + D)^{-1}, \\ \varphi^{(I)} \to [\det(C\tau + D)]^{k_I} \rho^{(I)}(\gamma)\varphi^{(I)}, \end{cases} \qquad \gamma = \begin{pmatrix} A & B \\ C & D \end{pmatrix} \in \Gamma_g. \tag{18}$$

In the case of rigid supersymmetry, the action $\mathcal{S}$ of an $\mathcal{N} = 1$ symplectic modular invariant supersymmetric theory, restricted to Yukawa interactions, reads

$$\mathcal{S} = \int d^4x \, d^2\theta \, d^2\bar{\theta} \, K(\Phi, \bar{\Phi}) + \int d^4x \, d^2\theta \, w(\Phi) + \text{h.c.}, \tag{19}$$

and its invariance under eq. (18) requires the invariance of the superpotential $w(\Phi)$ and the invariance of the Kahler potential up to a Kahler transformation:

$$\begin{cases} w(\Phi) \to w(\Phi) \\ K(\Phi, \bar{\Phi}) \to K(\Phi, \bar{\Phi}) + f(\Phi) + f(\bar{\Phi}) \end{cases}. \tag{20}$$

---

[2]We restrict to integer modular weights. Fractional weights are in general allowed, but require a suitable multiplier system [41, 73].

The invariance of the Kähler potential can be easily accomplished. A minimal Kähler potential is:

$$K = K_\tau + K_\varphi, \tag{21}$$

where:

$$K_\tau = -h \, \Lambda^2 \log \det(-i\tau + i\tau^\dagger), \qquad h > 0, \tag{22}$$

with $h$ a dimensionless constant and $\Lambda$ some reference mass scale. $K_\tau$ is invariant under the full symplectic group $Sp(2g, \mathbb{R})$ up to a Kähler transformation. The minimal Kähler potential $K_\varphi$ for matter multiplets $\varphi^{(I)}$ is invariant only under transformations of $\Gamma_g$:

$$K_\varphi = \sum_I [\det(-i\tau + i\tau^\dagger)]^{k_I} |\varphi^{(I)}|^2. \tag{23}$$

The requirement of symplectic modular invariance for the superpotential is better appreciated by expanding of $w(\Phi)$ in power series of the supermultiplets $\varphi^{(I)}$:

$$w(\Phi) = \sum_n Y_{I_1 \ldots I_n}(\tau) \, \varphi^{(I_1)} \ldots \varphi^{(I_n)}. \tag{24}$$

For the $p$-th order term to be modular invariant the functions $Y_{I_1 \ldots I_p}(\tau)$ should transform as Siegel modular forms with weight $k_Y(p)$ in the representation $\rho^{(Y)}$ of $\Gamma_{g,n}$:

$$Y_{I_1 \ldots I_p}(\gamma\tau) = [\det(C\tau + D)]^{k_Y(p)} \rho^{(Y)}(\gamma) \, Y_{I_1 \ldots I_p}(\tau), \tag{25}$$

with $k_Y(p)$ and $\rho^{(Y)}$ such that:

1. The weight $k_Y(p)$ should compensate the overall weight of the product $\varphi^{(I_1)} \ldots \varphi^{(I_p)}$:

$$k_Y(p) + k_{I_1} + \ldots + k_{I_p} = 0. \tag{26}$$

2. The product $\rho^{(Y)} \times \rho^{(I_1)} \times \ldots \times \rho^{(I_p)}$ contains an invariant singlet.

This framework can be easily extended to the case of local supersymmetry.

## 2.4 A constraint on modular transformations

Before discussing the inclusion of CP in this class of theories, we comment about a constraint applying to their transformation laws under the Siegel modular group $\Gamma_g$. The element $S^2 = -\mathbb{1}_{2g}$ commutes with all elements of $\Gamma_g$, therefore by Schur's Lemma the representation matrix $\rho_\mathbf{r}(S^2)$ of the finite modular group $\Gamma_{g,n}$ is proportional to a unit matrix, for any representation $\mathbf{r}$. Furthermore, we have $S^4 = \mathbb{1}_{2g}$ and $\rho_\mathbf{r}(S^4) = \mathbb{1}$, which implies $\rho_\mathbf{r}(S^2) = \pm\mathbb{1}$. Consider modular forms $Y(\tau)$ and matter multiplets $\varphi$ at genus $g$ and level $n$ transforming in the same irreducible representation $\rho_\mathbf{r}(\gamma)$ of $\Gamma_{g,n}$ under $\gamma \in \Gamma_g$:

$$Y(\gamma\tau) = [\det(C\tau + D)]^{k_Y} \rho_\mathbf{r}(\gamma) Y(\tau), \qquad \gamma = \begin{pmatrix} A & B \\ C & D \end{pmatrix} \in \Gamma_g, \tag{27}$$

$$\varphi \xrightarrow{\gamma} [\det(C\tau + D)]^{k_\varphi} \rho_\mathbf{r}(\gamma)\varphi. \tag{28}$$

By choosing $\gamma = S^2$ and observing that $Y(S^2\tau) = Y(\tau)$, we get:

$$Y(\tau) = [\det(-\mathbb{1}_g)]^{k_Y} \rho_\mathbf{r}(S^2) Y(\tau). \tag{29}$$

If $Y(\tau)$ does not vanish, this implies:

$$(-1)^{g k_Y} \rho_\mathbf{r}(S^2) = \mathbb{1}_\mathbf{r}. \tag{30}$$

Similarly, by considering twice the transformation $S^2$ on the matter multiplet $\varphi$, we get:

$$\varphi \xrightarrow{S^2} (-1)^{gk_\varphi}\rho_{\mathbf{r}}(S^2)\varphi \xrightarrow{S^2} [(-1)^{gk_\varphi}\rho_{\mathbf{r}}(S^2)]^2\varphi = \varphi\,, \tag{31}$$

and we find:

$$(-1)^{gk_\varphi}\rho_{\mathbf{r}}(S^2) = \pm\mathbb{1}_{\mathbf{r}}\,. \tag{32}$$

The matrix $\rho_{\mathbf{r}}(S^2)$ is independent from $k$, since it reflects a property of the group $\Gamma_{g,n}$. Therefore, for any genus $g$, eqs. (30) and (32) provide a set of constraints on the weights $k_Y$ and $k_\varphi$, once the representations $\rho_{\mathbf{r}}$ is chosen. It is instructive to analyze the mutual consistency of eqs. (30) and (32). If the genus $g$ is even, the only value of $\rho_{\mathbf{r}}(S^2)$ compatible with eq. (30) is $\rho_{\mathbf{r}}(S^2) = \mathbb{1}$. Let $\mathscr{R}_{MF}$ denote the set of all irreducible representations of $\Gamma_{g,n}$ in (27) for all possible integer values of the weight $k_Y$. This set might not coincide with $\mathscr{R}$, the set of all irreducible representations of $\Gamma_{g,n}$. If $\mathscr{R} - \mathscr{R}_{MF}$ is not empty, there are representations of $\Gamma_{g,n}$, to which matter fields can be assigned, such that $\rho_{\mathbf{r}}(S^2) = -\mathbb{1}$. If the genus $g$ is odd, eq. (30) and eq. (32) become $(-1)^{k_Y}\rho_{\mathbf{r}}(S^2) = \mathbb{1}$ and $(-1)^{k_\varphi}\rho_{\mathbf{r}}(S^2) = \pm\mathbb{1}$, respectively. We see that they can be compatible, provided the weights of the matter fields $k_\varphi$ satisfy: $(-1)^{k_\varphi} = \pm(-1)^{k_Y}$. For example, let us consider $g = 1$ and the level $n = 4$. The inhomogeneous finite modular group is isomorphic to $S_4$, which has five irreducible representations $\mathbf{1}$, $\mathbf{1}'$, $\mathbf{2}$, $\mathbf{3}$ and $\mathbf{3}'$. In the doublet representation $\rho_{\mathbf{2}}(S^2) = \mathbb{1}_{\mathbf{2}}$. Even weight for matter fields in the doublet representation are possible, if we choose the plus sign in eq. (32). However, odd weights for matter multiplets in the doublet representation are allowed as well, and eq. (32) is satisfied with the minus sign. On the contrary, only even weight modular forms can transform in the doublet representation.

As we shall see, the relations (30) and (32) play an important role in the definition of a consistent CP transformation law for matter multiplets and modular forms at genus $g = 1$ and $g = 2$.

# 3 Consistent CP transformations

In a theory invariant under $\Gamma_g = Sp(2g,\mathbb{Z})$, consistent CP transformations correspond to outer automorphism $u(\gamma)$ of $\Gamma_g$ [46,47]:

$$\mathcal{CP}\,\gamma\,\mathcal{CP}^{-1} = u(\gamma)\,. \tag{33}$$

Each automorphism $u(\gamma)$ of $\Gamma_g$ can be described by [74]:

$$u(\gamma) = \chi(\gamma)\,\mathcal{U}\,\gamma\,\mathcal{U}^{-1}\,, \qquad \mathcal{U} \in \Gamma_g^*\,, \tag{34}$$

where $\Gamma_g^* = GSp(2g,\mathbb{Z})$ denote the extended Siegel modular group, consisting of all integral matrices $\mathcal{U}$ satisfying $\mathcal{U}^t J \mathcal{U} = \pm J$. The map $\chi(\gamma)$, called character of the Siegel modular group, is a homomorphism of $\Gamma_g$ into $\{\pm 1\}$. The group $\Gamma_g^*$ is generated by $\{S, T_i, U\}$, where the matrix $U$, satisfying $U J U^{-1} = -J$, is defined as:

$$U = \begin{pmatrix} -\mathbb{1}_g & 0 \\ 0 & \mathbb{1}_g \end{pmatrix}. \tag{35}$$

The Siegel modular group $\Gamma_g$ is a subgroup of $\Gamma_g^*$ and each element $\mathcal{U}$ of the group $\Gamma_g^*$ not belonging to $\Gamma_g$ can be uniquely decomposed as:

$$\mathcal{U} = U\gamma'\,, \qquad \gamma' \in \Gamma_g\,. \tag{36}$$

Hence outer automorphisms $u(\gamma)$ of $\Gamma_g$ are recovered from eq. (34) by replacing the generic element $\mathcal{U}$ of $\Gamma_g^*$ either with the generator $U$ or with the identity:

$$u(\gamma) = \chi(\gamma) \, U \, \gamma \, U^{-1} \quad \text{and} \quad u(\gamma) = \chi(\gamma)\gamma. \tag{37}$$

Non-trivial characters $\chi(\gamma)$ exist only for $g = 1, 2$ [74, 75]:

   i) $g = 1$

$$\chi(\gamma) = \{1, \, \epsilon(\gamma)\}, \quad \text{where } \epsilon(S) = \epsilon(T) = -1. \tag{38}$$

   ii) $g = 2$

$$\chi(\gamma) = \{1, \, \theta(\gamma)\}, \quad \text{where } \theta(S) = 1, \quad \theta(T_i) = -1. \tag{39}$$

   iii) $g \geq 3$

$$\chi(\gamma) = 1. \tag{40}$$

For genus $g = 1, 2$ they give rise to two independent outer automorphisms $u_{1,2} = \chi_{1,2}(\gamma) \times U \, \gamma \, U^{-1}$, satisfying the following relations:

$$u_1^2 = u_2^2 = (u_1 u_2)^2 = \mathbb{1}. \tag{41}$$

Thus, the outer automorphism group is isomorphic to a Klein group $K_4 = \{u_1, u_2, u_3 = u_1 u_2, u_4 = \mathbb{1}\}$ [3]. For genus $g > 2$ the outer automorphism group is isomorphic to $Z_2 = \{u, u_4 = \mathbb{1}\}$. From eqs. (35) and (37) and the relation

$$U \, \gamma \, U^{-1} = \begin{pmatrix} A & -B \\ -C & D \end{pmatrix}, \tag{42}$$

we can derive the action of the outer automorphisms on the generators of $\Gamma_g$.

$$g = 1: \quad \begin{cases} u_1(S) = S^{-1}, & u_1(T) = T^{-1}, \\ u_2(S) = -S^{-1}, & u_2(T) = -T^{-1}, \\ u_3(S) = -S, & u_3(T) = -T, \\ u_4(S) = S, & u_4(T) = T, \end{cases} \tag{43}$$

$$g = 2: \quad \begin{cases} u_1(S) = S^{-1}, & u_1(T_i) = T_i^{-1}, \\ u_2(S) = S^{-1}, & u_2(T_i) = -T_i^{-1}, \\ u_3(S) = S, & u_3(T_i) = -T_i, \\ u_4(S) = S, & u_4(T_i) = T_i, \end{cases} \tag{44}$$

$$g \geq 3: \quad \begin{cases} u_1(S) = S^{-1}, & u_1(T_i) = T_i^{-1}, \\ u_4(S) = S, & u_4(T_i) = T_i. \end{cases} \tag{45}$$

We should select among these possibilities the good candidates to represent physical CP transformations. Notice that the automorphism $u$ is involutive for both $u = u_1$ and $u = u_2$: $u_i(u_i(\gamma)) = \gamma$ ($i = 1, 2$).

---

[3]We denote by $\mathbb{1}$ the identity element of the outer automorphism.

### 3.1 CP transformation of moduli $\tau$

We assume that the CP transformation of moduli $\tau$ is linear and, for convenience, we write [4]:

$$\tau \xrightarrow{\mathcal{CP}} (\mathcal{CP})\tau \equiv \tau_{\mathcal{CP}} = X \tau^* X^t, \tag{46}$$

where $X$ is an invertible $g \times g$ matrix such that $\mathrm{Im}(\tau_{\mathcal{CP}}) > 0$. The inverse CP transformation reads:

$$\tau \xrightarrow{\mathcal{CP}^{-1}} \tau_{\mathcal{CP}^{-1}} = (X^{-1} \tau (X^t)^{-1})^*. \tag{47}$$

By enforcing the relation (33) on the generators $S$ and $T_i$ of $\Gamma_g$, for the automorphisms $u_a(\gamma)$ $(a = 1, ..., 4)$, we get:

$$\begin{aligned} XX^t (-\tau^{-1}) XX^t &= u_a(S)\tau, \\ \tau + XB_iX^t &= u_a(T_i)\tau. \end{aligned} \tag{48}$$

The elements $u_1$ and $u_2$ have the same action on $\tau$ and similarly for $u_{3,4}$. As a consequence eq. (48) reduces to:

$$XX^t (-\tau^{-1}) XX^t = -\tau^{-1}, \qquad XB_iX^t = -\eta B_i, \tag{49}$$

where $\eta = +1$ for $u_{1,2}$ and $\eta = -1$ for $u_{3,4}$. Since $B_i$ form a set of basis of integral symmetric matrices, we see that the second set of relations is solved by $X = \pm i\mathbb{1}_g$ when $u_{1,2}$ and by $X = \pm\mathbb{1}_g$ for $u_{3,4}$. These solutions imply, respectively, $XX^t = -\mathbb{1}_g$ and $XX^t = \mathbb{1}_g$, both satisfying the first equation in (49). Since $\mathrm{Im}(\tau_{\mathcal{CP}}) > 0$, the only consistent choice is $X = \pm i\mathbb{1}_g$ and we find:

$$\tau \xrightarrow{\mathcal{CP}} \tau_{\mathcal{CP}} = -\tau^*, \tag{50}$$

as admissible CP transformation of the matrix $\tau$. This represents the correct transformation law of the moduli for both $u_{1,2}$ outer automorphisms. We should instead discard $u_{3,4}$. If we combine eq. (50) with a modular transformation:

$$\tau \xrightarrow{\gamma} (A\tau + B)(C\tau + D)^{-1} \xrightarrow{\mathcal{CP}} (\gamma \circ \mathcal{CP})\tau = (-A\tau^* + B)(-C\tau^* + D)^{-1}, \tag{51}$$

we get another allowed CP transformation. Since the theory is invariant under $\Gamma_g$, this choice should not be view as independent from (50), which we take as representative element in the class (51).

By combining CP and the Siegel modular transformations we get the extended Siegel modular group $\Gamma_g^* = GSp(2g, \mathbb{Z})$ and the full symmetry transformation of the complex moduli is

$$\begin{aligned} \tau &\to (A\tau + B)(C\tau + D)^{-1} \quad \text{for} \quad \gamma^t J\gamma = J, \\ \tau &\to (A\tau^* + B)(C\tau^* + D)^{-1} \quad \text{for} \quad \gamma^t J\gamma = -J, \end{aligned} \tag{52}$$

where $\gamma = \begin{pmatrix} A & B \\ C & D \end{pmatrix}$ and the CP transformation is represented by the matrix $U$ in eq. (35). Notice that the action of $\gamma$ and $-\gamma$ on $\tau$ is the same and the full symmetry group acting on moduli is isomorphic to $PGSp(2g, \mathbb{Z}) \equiv GSp(2g, \mathbb{Z})/\{\pm\mathbb{1}_{2g}\}$.

---

[4]We see at the end of this Section that also non-linear actions of CP are allowed. They are however equivalent to the linear one.

### 3.2 CP transformations of matter chiral multiplets $\varphi$

We consider a generic matter chiral supermultiplet $\varphi$, transforming as in eq. (28) under the Siegel modular group. We assume the following action of $CP$ on $\varphi$:

$$\varphi(x) \xrightarrow{\mathcal{CP}} X_{\mathbf{r}} \overline{\varphi}(x_{\mathcal{P}}), \tag{53}$$

where $X_{\mathbf{r}}$ is a unitary matrix, and a bar denotes hermitian conjugation. By realizing the condition (33) on the matter field space, we find:

$$[\det(C(\tau^*)_{\mathcal{CP}^{-1}} + D)]^{k_\varphi} X_{\mathbf{r}} \rho_{\mathbf{r}}^*(\gamma) X_{\mathbf{r}}^{-1} \varphi = \chi(\gamma)^{gk_\varphi} [\det(-C\tau + D)]^{k_\varphi} \rho_{\mathbf{r}}(u(\gamma)) \varphi, \tag{54}$$

where we made use of the relation:

$$u(\gamma) = \chi(\gamma) \begin{pmatrix} A & -B \\ -C & D \end{pmatrix}. \tag{55}$$

We conclude that:

$$X_{\mathbf{r}} \rho_{\mathbf{r}}^*(\gamma) X_{\mathbf{r}}^{-1} = \chi(\gamma)^{gk_\varphi} \rho_{\mathbf{r}}(u(\gamma)). \tag{56}$$

When the automorphism $u_1$ is chosen, from the previous equation we find:

$$X_{\mathbf{r}} \rho_{\mathbf{r}}^*(S) X_{\mathbf{r}}^{-1} = \rho_{\mathbf{r}}(S^{-1}), \qquad X_{\mathbf{r}} \rho_{\mathbf{r}}^*(T_i) X_{\mathbf{r}}^{-1} = \rho_{\mathbf{r}}(T_i^{-1}). \tag{57}$$

By making use of eqs. (38-39) and (43-44), for the automorphism $u_2$ we get

$$\chi_2(S)^{gk_\varphi} \rho_{\mathbf{r}}(u_2(S)) = [(-1)^{gk_\varphi} \rho_{\mathbf{r}}(S^2)]^g \rho_{\mathbf{r}}(S^{-1})$$
$$\chi_2(T_i)^{gk_\varphi} \rho_{\mathbf{r}}(u_2(T_i)) = [(-1)^{gk_\varphi} \rho_{\mathbf{r}}(S^2)] \rho_{\mathbf{r}}(T_i^{-1}). \tag{58}$$

We exploit the results of Section 2.4 and discuss separately the cases $g = 1$ and $g = 2$. When $g = 1$, the matrix $X_{\mathbf{r}}$ depends on both $\rho_{\mathbf{r}}(S^2)$ and $k_\varphi$ [5] and we distinguish two cases:

- $(-1)^{k_\varphi} \rho_{\mathbf{r}}(S^2) = +\mathbb{1}_{\mathbf{r}}$
  By combining eqs. (56) and (58), we get eq. (57) also for the automorphism $u_2$:

$$X_{\mathbf{r}} \rho_{\mathbf{r}}^*(S) X_{\mathbf{r}}^{-1} = \rho_{\mathbf{r}}(S^{-1}), \qquad X_{\mathbf{r}} \rho_{\mathbf{r}}^*(T_i) X_{\mathbf{r}}^{-1} = \rho_{\mathbf{r}}(T_i^{-1}). \tag{59}$$

- $(-1)^{k_\varphi} \rho_{\mathbf{r}}(S^2) = -\mathbb{1}_{\mathbf{r}}$
  In this case $X_{\mathbf{r}}$ obeys:

$$X_{\mathbf{r}} \rho_{\mathbf{r}}^*(S) X_{\mathbf{r}}^{-1} = -\rho_{\mathbf{r}}(S^{-1}), \quad X_{\mathbf{r}} \rho_{\mathbf{r}}^*(T) X_{\mathbf{r}}^{-1} = -\rho_{\mathbf{r}}(T^{-1}). \tag{60}$$

Note that each of these two conditions can be realized for any representation $\mathbf{r}$, with a suitable choice of $k_\varphi$.

On the contrary, when $g = 2$, the representations $\mathbf{r}$ of the finite modular group fall into two classes:

- $\rho_{\mathbf{r}}(S^2) = +\mathbb{1}_{\mathbf{r}}$
  By combining eqs. (56) and (58), we get again eq. (57):

$$X_{\mathbf{r}} \rho_{\mathbf{r}}^*(S) X_{\mathbf{r}}^{-1} = \rho_{\mathbf{r}}(S^{-1}), \qquad X_{\mathbf{r}} \rho_{\mathbf{r}}^*(T_i) X_{\mathbf{r}}^{-1} = \rho_{\mathbf{r}}(T_i^{-1}). \tag{61}$$

---

[5] We should more precisely denote $X_{\mathbf{r}}$ as $X_{\mathbf{r}}(k_\varphi)$, but in the text we leave the dependence on $k_\varphi$ understood.

- $\rho_{\mathbf{r}}(S^2) = -\mathbb{1}_{\mathbf{r}}$
  In this case $X_{\mathbf{r}}$ obeys:

$$X_{\mathbf{r}}\rho_{\mathbf{r}}^*(S)X_{\mathbf{r}}^{-1} = \rho_{\mathbf{r}}(S^{-1}), \quad X_{\mathbf{r}}\rho_{\mathbf{r}}^*(T)X_{\mathbf{r}}^{-1} = -\rho_{\mathbf{r}}(T^{-1}). \tag{62}$$

Now $X_{\mathbf{r}}$ is completely determined by $\rho_{\mathbf{r}}(S^2)$ and does not depend on $k_\varphi$.

The automorphism $u_2$ has been discussed in [50], for $g = 1$. The CP transformation defined by eqs. (60) and (62) can possibly be consistently implemented if:

1. The level $n$ is even, which follows from taking the $n$-th power of the second relation in eqs. (60, 62).

2. The dimension of representation $\rho_{\mathbf{r}}$ is even, which follows from eqs. (60, 62) by taking determinants.

3. The traces of $\rho_{\mathbf{r}}(T)$ and, for $g = 1$, $\rho_{\mathbf{r}}(S)$ should vanish, which follows from eqs. (60, 62) by taking traces.

It is not inconceivable to build a model with these properties [50]. We do not deal with such case here and, in the rest of our paper, we discuss the automorphism $u_2$, focusing on the solution $(-1)^{gk_\varphi}\rho_{\mathbf{r}}(S^2) = +\mathbb{1}_{\mathbf{r}}$. Then, both $u_1$ and $u_2$ satisfy eq. (57), that can be regarded as a set of consistency conditions on the unitary matrix $X_{\mathbf{r}}$. Indeed, multiplying from the right each member of the previous equations by $X_{\mathbf{r}}$, we get linear equations in the unknown $X_{\mathbf{r}}$, which admits a unique solution, up to an overall phase factor.

The action of $\mathcal{CP}$ in the moduli space, eq. (50), is involutive, that is $\mathcal{CP}^2\tau = \tau$. This is not necessarily the case in field space. From (33), by applying twice $\mathcal{CP}$, we get:

$$\mathcal{CP}^2 \, \gamma \, \mathcal{CP}^{-2} = u(u(\gamma)) \equiv \gamma \,, \tag{63}$$

showing that $\mathcal{CP}^2$ is an inner automorphism, which can be induced by an element $\gamma_{\mathcal{CP}^2}$ satisfying:

$$\gamma_{\mathcal{CP}^2} \, \gamma \, \gamma_{\mathcal{CP}^2}^{-1} = \gamma \,. \tag{64}$$

Therefore $\gamma_{\mathcal{CP}^2}$ belongs to the center of $\Gamma_g$: $\{\mathbb{1}_{2g}, S^2\}$. By realizing the equality (63) in the matter field space, we obtain:

$$\det(C\tau + D)^{k_\varphi}X_{\mathbf{r}}X_{\mathbf{r}}^* \, \rho_{\mathbf{r}}(\gamma) \, X_{\mathbf{r}}^{-1*}X_{\mathbf{r}}^{-1}\varphi = \det(C\tau + D)^{k_\varphi}\rho_{\mathbf{r}}(\gamma)\varphi \,. \tag{65}$$

By the Schur's Lemma, the product $X_{\mathbf{r}}X_{\mathbf{r}}^*$ is proportional to the identity. It acts without conjugating the matter fields and represents the element $\gamma_{\mathcal{CP}^2}$ or, more precisely, the element of the finite group $\Gamma_{g,n}$ that corresponds to $\gamma_{\mathcal{CP}^2}$. As a consequence we have:

$$X_{\mathbf{r}}X_{\mathbf{r}}^* = \mathbb{1}_{\mathbf{r}} \text{ or } X_{\mathbf{r}}X_{\mathbf{r}}^* = \rho_{\mathbf{r}}(S^2) = \pm\mathbb{1}_{\mathbf{r}}. \tag{66}$$

The latter equality follows from $S^4 = \mathbb{1}_{2g}$. Hence, the unitary matrix $X_{\mathbf{r}}$ is either symmetric or antisymmetric. The indicator $\text{Ind}_{\mathbf{r}} = \frac{1}{|\Gamma_{g,n}|}\sum_{\gamma \in \Gamma_{g,n}} \text{Tr}(\rho_{\mathbf{r}}(\gamma u_1(\gamma)))$ provides a criterion for deciding whether $X_{\mathbf{r}}X_{\mathbf{r}}^* = \mathbb{1}_{\mathbf{r}}$ or $X_{\mathbf{r}}X_{\mathbf{r}}^* = -\mathbb{1}_{\mathbf{r}}$ [47]. The indicator $\text{Ind}_{\mathbf{r}}$ is 1 and $-1$ for positive and negative sign respectively.

When $X_{\mathbf{r}}X_{\mathbf{r}}^* = \mathbb{1}_{\mathbf{r}}$, it is always possible to move to a basis where CP is canonical: $X_{\mathbf{r}} = \mathbb{1}_{\mathbf{r}}$. Then, the consistency conditions of eq. (57) imply that both $\rho_{\mathbf{r}}(S)$ and $\rho_{\mathbf{r}}(T)$ are symmetric in this basis: $\rho_{\mathbf{r}}^t(S) = \rho_{\mathbf{r}}(S)$ and $\rho_{\mathbf{r}}^t(T_i) = \rho_{\mathbf{r}}(T_i)$. Conversely, when $\rho_{\mathbf{r}}(S)$ and $\rho_{\mathbf{r}}(T)$ are symmetric, the relations (57) unify into [6]:

$$X_{\mathbf{r}} \, \rho_{\mathbf{r}}^*(\gamma) \, X_{\mathbf{r}}^{-1} = \rho_{\mathbf{r}}^t(\gamma^{-1}), \tag{67}$$

---

[6]It follows from $\gamma = S^{\alpha_1}\cdots T_i^{\beta_p}$ implying $\rho_{\mathbf{r}}^t(\gamma^{-1}) = \rho_{\mathbf{r}}^t(S^{-\alpha_1})\ldots\rho_{\mathbf{r}}^t(T_i^{-\beta_p}) = \rho_{\mathbf{r}}(S^{-\alpha_1})\ldots\rho_{\mathbf{r}}(T_i^{-\beta_p}) = \rho_{\mathbf{r}}(S^{-\alpha_1}\ldots T_i^{-\beta_p}) = \rho_{\mathbf{r}}(u_1(\gamma))$ in a symmetric basis.

which is solved by $X_{\mathbf{r}} = \mathbb{1}_{\mathbf{r}}$ and the action of CP is involutive also in the field space. We see that $u(\gamma)$ lies in the same conjugacy class as $\gamma^{-1}$ in the finite Siegel modular group. In other words $u(\gamma)$ is a class-inverting automorphism of the finite Siegel modular group $\Gamma_{g,n}$.

When $X_{\mathbf{r}}X_{\mathbf{r}}^* = \rho_{\mathbf{r}}(S^2) = -\mathbb{1}_{\mathbf{r}}$, $X_{\mathbf{r}}$ is a unitary antisymmetric matrix. In this case, the dimension of the representation $\rho_{\mathbf{r}}$ has to be even. By performing a field redefinition we can go to a basis where $X_{\mathbf{r}}$ takes the form [76]

$$X_{\mathbf{r}} = \begin{pmatrix} i\sigma_2 & & \\ & \ddots & \\ & & i\sigma_2 \end{pmatrix}, \tag{68}$$

where $\sigma_2$ is the second Pauli matrix. Examples of non-involutive CP transformations on the field space have been given in ref. [77]. In most of our discussion, we focus on the involutive case.

### 3.3 CP transformations of modular forms $Y(\tau)$

Consider a multiplet $Y(\tau)$ of modular forms at genus $g$, level $n$ and weight $k_Y$, transforming as in eq. (27) under the Siegel modular group. In general, there can be several linearly independent such multiplets. We start by examining the case where there is only one. Under CP it transforms as:

$$Y(\tau) \xrightarrow{\mathcal{CP}} Y(-\tau^*). \tag{69}$$

We would like to establish the relation between $Y(-\tau^*)$ and $X_{\mathbf{r}}Y^*(\tau)$, where $X_{\mathbf{r}}$ is the matrix specifying the transformation law under CP of a matter multiplet $\varphi$ at the same genus $g$, level $n$, weight $k_\varphi = k_Y$ and irreducible representation $\rho_{\mathbf{r}}(\gamma)$, see eqs. (28) and (53). Note that in this case the matrix $X_{\mathbf{r}}$ satisfies necessarily the consistency condition associated with $(-1)^{g k_\varphi} \rho_{\mathbf{r}}(S^2) = +\mathbb{1}_{\mathbf{r}}$. Indeed the other sign choice is only possible for $\mathbf{r}$ belonging to $\mathscr{R} - \mathscr{R}_{MF}$, when $g$ is even and $k_\varphi \neq k_Y$, when $g$ is odd. Thus, among all possible matrices $X_{\mathbf{r}}$, here the only relevant ones are the solution of the consistency conditions (57). We define:

$$\widetilde{Y}(\tau) = X_{\mathbf{r}}^{-1*} Y^*(-\tau^*). \tag{70}$$

We see that under $\gamma \in \Gamma_g$, $\widetilde{Y}(\tau)$ transforms as:

$$\begin{aligned}
\widetilde{Y}(\gamma\tau) &= X_{\mathbf{r}}^{-1*} Y^*(-(\gamma\tau)^*) \\
&= X_{\mathbf{r}}^{-1*} Y^*(u(\gamma)(-\tau^*)) \\
&= X_{\mathbf{r}}^{-1*} [\det(C\tau+D)]^k \chi(\gamma)^{gk} \rho_{\mathbf{r}}^*(u(\gamma)) \, Y^*(-\tau^*) \\
&= [\det(C\tau+D)]^k \, \rho_{\mathbf{r}}(\gamma) \, \widetilde{Y}(\tau),
\end{aligned} \tag{71}$$

where, in the last equality, we have used eq. (56). Since $\widetilde{Y}(\tau)$ and $Y(\tau)$ transform in the same way and, by assumption, there is only one linearly independent such modular form, we conclude that they are proportional, that is:

$$Y(-\tau^*) = \lambda \, X_{\mathbf{r}} Y^*(\tau). \tag{72}$$

By performing an additional CP transformation we have $Y(\tau) = |\lambda|^2 X_{\mathbf{r}} X_{\mathbf{r}}^* Y(\tau) = |\lambda|^2 Y(\tau)$, where we have made use of eq. (66). Note that only the solution $X_{\mathbf{r}} X_{\mathbf{r}}^* = +\mathbb{1}_{\mathbf{r}}$ applies in this case. The non vanishing constant $\lambda$ can be absorbed by an appropriate choice of phase of the whole multiplet $Y(\tau)$. In such a basis of modular forms $Y(\tau)$ we have:

$$Y(-\tau^*) = X_{\mathbf{r}} Y^*(\tau), \tag{73}$$

which reproduces the same CP transformation law of matter fields.

If there are $N$ linearly independent multiplets $Y^a(\tau)$ ($a = 1, ..., N$) transforming as in (27), eq. (71) holds individually for all $\widetilde{Y}^a(\tau) = X_{\mathbf{r}}^{-1*}Y^{a*}(-\tau^*)$ and we have:

$$Y^a(-\tau^*) = \lambda^a_{\ b} X_{\mathbf{r}}Y^{b*}(\tau), \tag{74}$$

where $\lambda^a_{\ c}\lambda^{*c}_{\ b}X_{\mathbf{r}}X_{\mathbf{r}}^* = \delta^a_{\ b}\mathbb{1}_{\mathbf{r}}$. From eq. (66), now we can only deduce $\lambda^a_{\ c}\lambda^{*c}_{\ b} = \pm\delta^a_{\ b}$, the sign plus (minus) applying when the action of $X_{\mathbf{r}}$ is (is not) involutive. When $X_{\mathbf{r}}$ is involutive and $\lambda^a_{\ c}\lambda^{*c}_{\ b} = \delta^a_{\ b}$, it is always possible to factorize the matrix $\lambda$ into $\lambda = \eta^{-1}\eta^*$ and we obtain[7]:

$$\eta^a_{\ b}Y^b(-\tau^*) = X_{\mathbf{r}}[\eta^a_{\ b}Y^b]^*(\tau). \tag{75}$$

We see that, by performing the change of basis $Y^a(\tau) \to \eta^a_{\ b}Y^b(\tau)$, eq. (73) holds independently for each multiplet $Y^a(\tau)$. In applications we will use such basis of modular forms.

The linear space $\mathcal{M}_k(\Gamma_g(n))$ of weight $k$ modular forms for $\Gamma_g(n)$ is finite dimensional and decomposes into the sum of invariant subspaces, each carrying an irreducible representation $(\rho^{a_I}_{\mathbf{r}_I})_{i_I j_I}$ of $\Gamma_{g,n}$ of dimension $d_I$. Here $a_I$ is an index describing the degeneracy of the representation $\rho_{\mathbf{r}_I}$ and the indices $(i_I, j_I)$ run from 1 to $d_I$. Let $\{(F^{a_I}_{\mathbf{r}_I})_{i_I}(\tau)\}$ denote a basis in $\mathcal{M}_k(\Gamma_g(n))$. Our result (74) implies:

$$(F^{a_I}_{\mathbf{r}_I})_{i_I}(-\tau^*) = \delta^I_{\ J}\lambda^{a_I}_{\ b_J}(X_{\mathbf{r}_J})^{j_I}_{\ i_I}(F^{b_J}_{\mathbf{r}_J})^*_{j_I}(\tau), \tag{76}$$

or, omitting indices,

$$F(-\tau^*) = X_F F^*(\tau). \tag{77}$$

From the properties of $\lambda^a_{\ b}$ and $X_{\mathbf{r}}$, it follows that

$$X_F X_F^* = \mathbb{1}, \qquad X_F \rho^*(\gamma) X_F^{-1} = \rho(u_1(\gamma)), \tag{78}$$

where $\rho(\gamma)$ is the direct sum of the representations $\{\rho^{a_I}_{\mathbf{r}_I}\}$.

Actually, we could reverse the logic of this section and start by proving eqs. (77) and (78) and finally conclude that eq. (74) should hold. Indeed, for any element $\gamma$ of $\Gamma_g(n)$, a basis $\{F(\tau)\}$ of $\mathcal{M}_k(\Gamma_g(n))$ obeys

$$F^*(-(\gamma\tau)^*) = F^*([A(-\tau^*) - B][-C(-\tau^*) + D]^{-1}) = [\det(C\tau + D)]^{k_Y} F^*(-\tau^*), \tag{79}$$

showing that $F^*(-\tau^*)$ belongs to $\mathcal{M}_k(\Gamma_g(n))$, which is the content of eq. (77). As a consequence, the following relation should be fulfilled:

$$F(\tau) = [F^*(-(-\tau^*)^*)]^* = [X_F^* F(-\tau^*)]^* = X_F X_F^* F(\tau), \tag{80}$$

which leads to $X_F X_F^* = \mathbb{1}$. Finally, by performing a Siegel modular transformation $\gamma \in \Gamma_g$ on both sides of eq. (77) and using the identity $-(\gamma\tau)^* = u(\gamma)(-\tau^*)$, we obtain

$$X_F \rho^*(\gamma) X_F^{-1} = \chi(\gamma)^{g k_Y} \rho(u(\gamma)) = \rho(u_1(\gamma)), \tag{81}$$

where $\rho(\gamma) = \mathrm{diag}(\rho^1_{\mathbf{r}_1}(\gamma), \ldots, \rho^{m1}_{\mathbf{r}_1}(\gamma), \ldots, \rho^1_{\mathbf{r}_p}(\gamma), \ldots, \rho^{mp}_{\mathbf{r}_p}(\gamma))$ is generally reducible. Then eq. (74) follows by projecting (77) on invariant subspaces.

---

[7]If $(\mathbb{1} + \lambda)$ is invertible, we take $\eta = (\mathbb{1} + \lambda)^{-1}$, then $\lambda\eta^{*-1} = \eta^{-1}$. If $(\mathbb{1} + \lambda)$ is not invertible, we can always find a complex number $u$ with $|u| = 1$ such that $-u^2$ is not an eigenvalue of $\lambda$. Hence, $\lambda + u^2\mathbb{1}$ is invertible. In this case, we take $\eta = (u^{-1}\lambda + u\mathbb{1})^{-1}$, then $\lambda\eta^{*-1} = \eta^{-1}$. This construction was given by Prof. Marc van Leeuwen [78].

### 3.4 Condition for CP invariance

We have seen that the transformation properties of moduli, matter multiplets and modular forms are given by:

$$
\begin{cases}
\tau \xrightarrow{\mathcal{CP}} -\tau^*, \\
\varphi(x) \xrightarrow{\mathcal{CP}} X_{\mathbf{r}} \overline{\varphi}(x_{\mathcal{P}}), \\
Y^a(\tau) \xrightarrow{\mathcal{CP}} Y^a(-\tau^*) = \lambda^a{}_b X_{\mathbf{r}} Y^{b*}(\tau),
\end{cases}
\tag{82}
$$

with $\lambda^a{}_c \lambda^{*c}{}_b X_{\mathbf{r}} X_{\mathbf{r}}^* = \delta^a{}_b \mathbb{1}_{\mathbf{r}}$. For both automorphism $u_1$ and the positive branch $(-1)^{g k_\varphi} \rho_{\mathbf{r}}(S^2) = +\mathbb{1}_{\mathbf{r}}$ of $u_2$, the unitary matrix $X_{\mathbf{r}}$ solves the consistency conditions (57) and is determined up to an arbitrary phase. When the action of CP on the field space is involutive, it is convenient to move to the basis where $X_{\mathbf{r}} = \mathbb{1}$. Here the matrices $\rho_{\mathbf{r}}(S)$ and $\rho_{\mathbf{r}}(T)$ are symmetric and it is possible to work with modular forms $Y^a(\tau)$ where $\lambda^a{}_b = \delta^a{}_b$. The minimal Kähler potential, eqs. (22) and (23), is always invariant under CP. For the superpotential, the condition for CP invariance simplifies when *i)* $X_{\mathbf{r}} = \mathbb{1}$, *ii)* the Clebsh-Gordan coefficients are all real in the adopted basis and *iii)* the modular forms $Y(\tau)$ are in a basis where (73) holds. In this case the superpotential is CP invariant when its free parameters are real.

## 4 Points of residual CP symmetry

There are points of the moduli space $\mathcal{H}_g$ where CP is conserved. The theory is invariant under $\Gamma_g$. Moreover the inequalities (11) defining the fundamental domain $\mathcal{F}_g$ do not change when we map $\tau$ into $-\tau^*$. Thus it is sufficient to look for the CP invariant points belonging to $\mathcal{F}_g$. If $\tau$ belongs to the interior of $\mathcal{F}_g$, the CP invariant points are the solutions of:

$$
-\tau^* = \tau, \tag{83}
$$

that is the moduli $\tau$ of the fundamental domain with vanishing real part. Points of the boundary $\partial \mathcal{F}_g$ of $\mathcal{F}_g$, where at least one of the relations in eq. (11) is realized as an equality, are related by modular transformations. The requirement of CP conservation for any point $\tau$ of the boundary is the existence of a modular transformation $\gamma$, such that:

$$
-\tau^* = \gamma \tau. \tag{84}
$$

Indeed the composition of a CP transformation with a modular transformation is an equivalent CP transformation in our theory. By applying two consecutive such combinations, we obtain the condition:

$$
u(\gamma) \gamma \tau = \tau. \tag{85}
$$

We make use of the identity [72]

$$
(C\tau^* + D)^t \, \mathrm{Im}(\gamma \tau)(C\tau + D) = \mathrm{Im}(\tau). \tag{86}
$$

If $\tau$ satisfies eq. (84), we have $\mathrm{Im}(\gamma \tau) = \mathrm{Im}(-\tau^*) = \mathrm{Im}(\tau)$ and

$$
(C\tau^* + D)^t \, \mathrm{Im}(\tau)(C\tau + D) = \mathrm{Im}(\tau). \tag{87}
$$

By taking the determinant of both sides and noticing $\mathrm{Im}(\tau) > 0$, we get

$$
|\det(C\tau + D)| = 1. \tag{88}
$$

By comparing eq. (88) with eq. (11), we see that indeed $\tau$ belongs to the boundary $\partial \mathcal{F}_g$. We can distinguish two cases where eq. (88) is satisfied: $C = 0$ and $C \neq 0$. The elements of $\Gamma_g$ having $C = 0$ are of the type:

$$\gamma = \begin{pmatrix} A & M \\ 0 & A^{t-1} \end{pmatrix}, \tag{89}$$

where $A$ is an unimodular [8] integral matrix and $M$ is a symmetric modular matrix. In this case the relation (84) becomes:

$$-\text{Re}(\tau) = A(\text{Re}(\tau) + M)A^t$$
$$\text{Im}(\tau) = A\,\text{Im}(\tau)A^t. \tag{90}$$

If we consider $A = \mathbb{1}_g$, we get: $2\text{Re}(\tau) + M = 0$. Since $|\text{Re}(\tau)| \leq 1/2$, this relation is solved by:

$$\text{Re}(\tau) = 0, \quad \text{and} \quad \text{Re}(\tau_{ij}) = \pm\frac{1}{2}, \tag{91}$$

for $M = \mathbb{0}_g$ and $M = \pm B_i$ respectively. If $C \neq 0$, we consider as an example the choice $\gamma = S$. The condition eq. (84) becomes:

$$\tau^* = \tau^{-1}. \tag{92}$$

For genus $g = 1$, this is the arc $|\tau|^2 = 1$. In the case of $g = 2$, the CP conserved values of $\tau$ satisfy the conditions $|\tau_1|^2 + |\tau_3|^2 = |\tau_2|^2 + |\tau_3|^2 = 1$, $\tau_1 \tau_3^* + \tau_2^* \tau_3 = 0$, which are $|\tau_1|^2 = |\tau_2|^2 = 1$, $\tau_3 = 0$ in the fundamental domain $\mathcal{F}_2$. We might ask whether all points of the boundary $\partial \mathcal{F}_g$ ($g \geq 2$) enjoy CP invariance, as is the case for genus one. All points of $\partial \mathcal{F}_g$ satisfying $|\text{Re}(\tau_{ij})| = 1/2$, for at least one pair $(ij)$, are easily shown to be CP conserving, since the relation (84) is satisfied by a translation $\gamma$. For points satisfying the condition $|\det(C\tau + D)| = 1$ with $C \neq 0$, it is more difficult to establish, in general, if they are CP conserving or not. Showing this amounts to prove that, for a generic $\tau$ satisfying $|\det(C\tau + D)| = 1$ with $C \neq 0$, we can always determine a modular transformation $\gamma$ such that eq. (84) holds. To our best knowledge this is still an open problem.

In summary, in the interior of the $\mathcal{F}_g$, a point $\tau$ is CP conserving if and only if its real part is vanishing. There are additional CP conserving points on the boundary of $\mathcal{F}_g$, but we were not able to prove that any point belonging to $\partial \mathcal{F}_g$ is CP conserving.

## 4.1 Implications of residual CP symmetry

Consider a point $\tau$ of the fundamental domain where CP is conserved. Then there is an element $\gamma$ of $\Gamma_g$ such that $-\tau^* = \gamma\tau$. As we have seen, eq. (88), the combination

$$\mathscr{D} = \det(C\tau + D) \tag{93}$$

is a pure phase. Assume that the lepton sector is described by the superpotential:

$$\mathcal{W} = -E_i^c \mathcal{Y}_{ij}^e(\tau) L_j H_d - \frac{1}{2\Lambda} L_i \mathcal{Y}_{ij}^\nu(\tau) L_j H_u H_u, \tag{94}$$

where under the group element $\gamma$ the matter multiplets $\varphi$ ($\varphi = H_{u,d}, E^c, L$) transform as:

$$\varphi \xrightarrow{\gamma} \mathscr{D}^{k_\varphi} \rho_\varphi(\gamma)\varphi. \tag{95}$$

---

[8]($|\det A| = 1$).

The weights $k_{E^c,L}$ carry a flavour index and $\mathscr{D}^{k_{E^c,L}}$ are diagonal unitary matrices in flavour space. The invariance of the $\mathcal{W}$ under the modular transformation $\gamma$ implies:

$$\mathcal{Y}^e(\gamma\tau) = \mathscr{D}^{-k_{H_d}} \rho^\dagger_{H_d}(\gamma)\, \mathscr{D}^{-k_{E^c}} \rho^*_{E^c}(\gamma)\, \mathcal{Y}^e(\tau)\, \rho^\dagger_L(\gamma) \mathscr{D}^{-k_L},$$
$$\mathcal{Y}^\nu(\gamma\tau) = \mathscr{D}^{-2k_{H_u}} \rho^{2\dagger}_{H_u}(\gamma)\, \mathscr{D}^{-k_L} \rho^*_L(\gamma)\, \mathcal{Y}^\nu(\tau)\, \rho^\dagger_L(\gamma) \mathscr{D}^{-k_L}. \tag{96}$$

On the other hand, the invariance of $\mathcal{W}$ under a CP transformation requires:

$$\mathcal{Y}^e(-\tau^*) = X^\dagger_{H_d}\, X^*_{E^c}\, \mathcal{Y}^{e*}(\tau)\, X^\dagger_L,$$
$$\mathcal{Y}^\nu(-\tau^*) = X^{2\dagger}_{H_u}\, X^*_L\, \mathcal{Y}^{\nu*}(\tau)\, X^\dagger_L. \tag{97}$$

By combining eqs. (96) and (97), at the point $\tau$ enjoying residual CP symmetry we get:

$$\mathcal{Y}^{e*}(\tau) = \Omega_{H_d}\, \Omega^T_{E^c}\, \mathcal{Y}^e(\tau)\, \Omega_L,$$
$$\mathcal{Y}^{\nu*}(\tau) = \Omega^2_{H_u}\, \Omega^T_L\, \mathcal{Y}^\nu(\tau)\, \Omega_L, \tag{98}$$

where we have defined unitary matrices:

$$\Omega_\varphi = \rho^\dagger_\varphi(\gamma)\mathscr{D}^{-k_\varphi}X_\varphi, \qquad (\varphi = H_{u,d}, E^c, L). \tag{99}$$

The charged lepton and neutrino mass matrices are given by $M_e = \mathcal{Y}^e v_d$ and $M_\nu = \mathcal{Y}^\nu v_u^2/\Lambda$ for the minimal Kähler potential (23). Thus at the point $\tau$ enjoying residual CP invariance we have

$$\Omega^\dagger_L M^\dagger_e(\tau)M_e(\tau)\Omega_L = \left[M^\dagger_e(\tau)M_e(\tau)\right]^*,$$
$$\Omega^\dagger_L M^\dagger_\nu(\tau)M_\nu(\tau)\Omega_L = \left[M^\dagger_\nu(\tau)M_\nu(\tau)\right]^*. \tag{100}$$

We see that the hermitian combination of the neutrino and charged lepton mass matrices $M^\dagger_\nu M_\nu$ and $M^\dagger_e M_e$ are invariant under a common transformation of the left-handed charged leptons and the left-handed neutrinos, which represents a combination of CP and modular transformations. The unitary matrix $\Omega_L$ should be symmetric otherwise the neutrino and the charged lepton mass spectrum would be constrained to be partially degenerate [45]. Furthermore, the conditions eq. (100) fulfilled at the CP fixed point imply that both Dirac and Majorana CP phases are trivial [45]. Therefore values of moduli deviating from residual CP symmetry fixed points are required to accommodate the observed non-degenerate lepton masses and a non-trivial Dirac CP.

## 5 CP action in invariant subspaces

The supersymmetric modular invariant theory of Section 2 can be consistently defined even when $\tau$ belongs to an invariant subspace $\Omega$ of the Siegel moduli space $\mathcal{H}_g$. In this case the full Siegel modular group $\Gamma_g$ is replaced by a convenient subgroup $N(H)$. The points $\tau$ in $\Omega$ are fixed points of the subgroup $H$ of $\Gamma_g$ [79–81], while $N(H)$, the normalizer of $H$, leaves $\Omega$ invariant as a whole. The elements $\gamma$ of $N(H)$ satisfy the relation $\gamma^{-1}H\gamma = H$. Depending on $H$, the complex dimensionality of the invariant subspace $\Omega$ can range from zero to $g(g+1)/2$.

Under these conditions, eq. (50) might not be suitable to describe the action of CP on $\tau$, since we are not guaranteed that $-\tau^*$ belongs to $\Omega$. To remedy this situation, we can adopt the definition of CP given in eq. (51), where a modular transformation $\gamma$ is followed by the canonical CP transformation

$$\tau \to (\gamma \circ \mathcal{CP})\tau = (-A\tau^* + B)(-C\tau^* + D)^{-1}, \tag{101}$$

Table 1: The generalized CP transformation in the modular subspace in Siegel upper half plane $\mathcal{H}_2$. The complex moduli are denoted by $\tau_1, \tau_2, \tau_3$, and $\zeta = e^{2\pi i/5}$, $\eta = \frac{1}{3}(1 + i2\sqrt{2})$, $\omega = e^{2\pi i/3}$.

| Fixed points $\tau$ | $\mathcal{CP}_s$ | Fixed points $\tau$ | $\mathcal{CP}_s$ |
|---|---|---|---|
| $\begin{pmatrix} \tau_1 & 0 \\ 0 & \tau_2 \end{pmatrix}$ | $\mathcal{CP}$ | $\begin{pmatrix} \tau_1 & \tau_3 \\ \tau_3 & \tau_1 \end{pmatrix}$ | $\mathcal{CP}$ |
| $\begin{pmatrix} i & 0 \\ 0 & \tau_2 \end{pmatrix}$ | $\mathcal{CP}$ | $\begin{pmatrix} \omega & 0 \\ 0 & \tau_2 \end{pmatrix}$ | $T_1^{-1} \circ \mathcal{CP}$ |
| $\begin{pmatrix} \tau_1 & 0 \\ 0 & \tau_1 \end{pmatrix}$ | $\mathcal{CP}$ | $\begin{pmatrix} \tau_1 & 1/2 \\ 1/2 & \tau_1 \end{pmatrix}$ | $T_3 \circ \mathcal{CP}$ |
| $\begin{pmatrix} \tau_1 & \tau_1/2 \\ \tau_1/2 & \tau_1 \end{pmatrix}$ | $\mathcal{CP}$ | $\begin{pmatrix} \zeta & \zeta + \zeta^{-2} \\ \zeta + \zeta^{-2} & -\zeta^{-1} \end{pmatrix}$ | $((ST_3)^3 T_3)^{-1} \circ \mathcal{CP}$ |
| $\begin{pmatrix} \eta & \frac{1}{2}(\eta - 1) \\ \frac{1}{2}(\eta - 1) & \eta \end{pmatrix}$ | $ST_3(T_1 T_2)^{-1} S \circ \mathcal{CP}$ | $\begin{pmatrix} i & 0 \\ 0 & i \end{pmatrix}$ | $\mathcal{CP}$ |
| $\begin{pmatrix} \omega & 0 \\ 0 & \omega \end{pmatrix}$ | $(T_1 T_2)^{-1} \circ \mathcal{CP}$ | $\dfrac{i\sqrt{3}}{3} \begin{pmatrix} 2 & 1 \\ 1 & 2 \end{pmatrix}$ | $\mathcal{CP}$ |
| $\begin{pmatrix} \omega & 0 \\ 0 & i \end{pmatrix}$ | $T_1^{-1} \circ \mathcal{CP}$ | — | |

and require for any $\tau \in \Omega$:

$$(\gamma \circ \mathcal{CP})\tau = \tau' \in \Omega. \tag{102}$$

Since each point of $\Omega$ is fixed under $H$, the transformation $(\gamma \circ \mathcal{CP})$ should obey:

$$(\gamma \circ \mathcal{CP}) \circ H = H \circ (\gamma \circ \mathcal{CP}). \tag{103}$$

This condition determines $\gamma$ up to a modular transformation $\gamma'$ of the normalizer $N(H)$, given that it is satisfied by both $(\gamma \circ \mathcal{CP})$ and $(\gamma' \gamma \circ \mathcal{CP})$. Therefore, it is sufficient to choose a representative CP transformation in this class, and we denote it as $\mathcal{CP}_s$. For the case of $g = 2$, all independent invariant subspaces and the corresponding CP transformations are summarized in table 1, where all $\mathcal{CP}_s$ are chosen to be involutive with $(\mathcal{CP}_s^2)\tau = \tau$. There are 6 zero-dimensional invariant subspaces. The unique point belonging to each of these regions is CP invariant. In the modular subspaces of complex dimension one and two there are infinite CP-conserving points. We list below the CP-conserving points satisfying $(\mathcal{CP}_s)\tau = g_i^{-1}\tau$, where $g_i$ stand for the generators of $N(H)$. This request can be equivalently cast in the form[9]:

$$(g_i \circ \mathcal{CP}_s)\tau = \tau. \tag{104}$$

1. $\begin{pmatrix} \tau_1 & 0 \\ 0 & \tau_2 \end{pmatrix}$: $\begin{cases} G_1 \circ \mathcal{CP}_s : \ \mathrm{Re}(\tau_1) = 0, |\tau_2| = 1, \\ G_1' \circ \mathcal{CP}_s : \ \mathrm{Re}(\tau_2) = 0, |\tau_1| = 1, \\ G_2 \circ \mathcal{CP}_s : \ |\mathrm{Re}(\tau_1)| = 1/2, \mathrm{Re}(\tau_2) = 0, \\ G_2' \circ \mathcal{CP}_s : \ |\mathrm{Re}(\tau_2)| = 1/2, \mathrm{Re}(\tau_1) = 0, \\ G_3 \circ \mathcal{CP}_s : \ \mathrm{Re}(\tau_1) = -\mathrm{Re}(\tau_2), \mathrm{Im}(\tau_1) = \mathrm{Im}(\tau_2). \end{cases}$

---

[9]Given any CP conserving point $\tau$ of $g_i \circ \mathcal{CP}_s$, $(\mathcal{CP}_s)\tau$ would be the fixed point of $g_i^{-1} \circ \mathcal{CP}_s$.

2. $\begin{pmatrix} \tau_1 & \tau_3 \\ \tau_3 & \tau_1 \end{pmatrix}$: $\begin{cases} G_1 \circ \mathcal{CP}_s : & |\mathrm{Re}(\tau_1)| = 1/2, \mathrm{Re}(\tau_3) = 0, \\ G_2 \circ \mathcal{CP}_s : & |\mathrm{Re}(\tau_3)| = 1/2, \mathrm{Re}(\tau_1) = 0, \\ G_3 \circ \mathcal{CP}_s : & |\tau_1| = 1, \tau_3 = 0, \\ G_4 \circ \mathcal{CP}_s : & \mathrm{Re}(\tau_1) = \mathrm{Im}(\tau_3) = 0. \end{cases}$

3. $\begin{pmatrix} i & 0 \\ 0 & \tau_2 \end{pmatrix}$: $\begin{cases} R \circ \mathcal{CP}_s : & \mathrm{Re}(\tau_2) = 0, \\ G_1 \circ \mathcal{CP}_s : & |\tau_2| = 1, \\ G_2 \circ \mathcal{CP}_s : & |\mathrm{Re}(\tau_2)| = 1/2. \end{cases}$

4. $\begin{pmatrix} \omega & 0 \\ 0 & \tau_2 \end{pmatrix}$: $\begin{cases} R \circ \mathcal{CP}_s : & \mathrm{Re}(\tau_2) = 0, \\ G_1 \circ \mathcal{CP}_s : & |\tau_2| = 1, \\ G_2 \circ \mathcal{CP}_s : & |\mathrm{Re}(\tau_2)| = 1/2. \end{cases}$

5. $\begin{pmatrix} \tau_1 & 0 \\ 0 & \tau_1 \end{pmatrix}$: $\begin{cases} R_1 \circ \mathcal{CP}_s : & \mathrm{Re}(\tau_1) = 0, \\ G_1 \circ \mathcal{CP}_s : & |\tau_1| = 1, \\ G_2 \circ \mathcal{CP}_s : & |\mathrm{Re}(\tau_1)| = 1/2. \end{cases}$

6. $\begin{pmatrix} \tau_1 & 1/2 \\ 1/2 & \tau_1 \end{pmatrix}$: $\begin{cases} G_1 \circ \mathcal{CP}_s : & \mathrm{Re}(\tau_1) = 0, \\ R \circ \mathcal{CP}_s : & |\mathrm{Re}(\tau_1)| = 1/2, \\ G_2 R \circ \mathcal{CP}_s : & (\mathrm{Re}(\tau_1) \pm 1/2)^2 + (\mathrm{Im}(\tau_1))^2 = 1/2. \end{cases}$

7. $\begin{pmatrix} \tau_1 & \tau_1/2 \\ \tau_1/2 & \tau_1 \end{pmatrix}$: $\begin{cases} G_1 \circ \mathcal{CP}_s : & |\tau_1| = 2/\sqrt{3}, \\ G_2 \circ \mathcal{CP}_s : & (\mathrm{Re}(\tau_1) \pm 2/3)^2 + (\mathrm{Im}(\tau_1))^2 = 4/9, \\ G_2' \circ \mathcal{CP}_s : & \mathrm{Re}(\tau_1) = 0. \end{cases}$

Here $G_1, G_2, G_1', G_2', R$ in each case are the generators of $N(H)$ associated with the two bidimensional and the five unidimensional invariant subspaces shown in table 1. Their matrix representation in a convenient basis is given in ref. [2].

# 6 A model with CP invariance at genus 2

In a CP-invariant theory, CP violation can arise only as a consequence of the choice of the vacuum. If in addition the theory enjoys flavour modular invariance, the properties of the observed fermion spectrum such as masses, mixing angles and phases could be determined mostly by the vacuum, rather than by Lagrangian parameters. These features are parts of an appealing framework for the unification of flavor, CP and modular symmetries, advocated in recent works [39–44] in a top-down perspective. In a bottom-up approach we can hope to explore some aspect of this ideal framework.

In this spirit, here we present an example of how CP and flavour symmetries can be combined and enforced in a supersymmetric theory, where the properties of the mass spectrum depend in a non-trivial way on a multidimensional moduli space. When a single modulus exists, analogous studies have been carried out in refs. [48, 50–53]. We focus on the lepton sector, where we have 12 relevant observables. We show that all measured mass combinations and mixings can be correctly described in terms of five Lagrangian parameters and by a convenient point in moduli space [10]. The theory is CP invariant and all 3 CP violating phases arise

---

[10]Note that the most economic modular-invariant flavour models describe the lepton sector by making use of five parameters.

spontaneously as a consequence of a small departure of $\tau$ from a CP-symmetric point.

It is convenient to base our construction on the invariant subspace $\tau_1 = \tau_2$ at genus $g = 2$ [2]. At level 2, the modular group $N(H)$ [11] is $S_4 \times Z_2$, whose generators $G_1 = T_1 T_2$, $G_2 = T_3$ and $G_3 = S$ are shown in appendix A, in a convenient basis. We adopt $u(\gamma) = u_1(\gamma)$ as the automorphism defining CP. In terms of $G_i$ ($i = 1, 2, 3$), the consistency conditions of eq. (57) become:

$$X_{\mathbf{r}} \rho_{\mathbf{r}}^*(G_i) X_{\mathbf{r}}^{-1} = \rho_{\mathbf{r}}(u(G_i)) = \rho_{\mathbf{r}}(G_i^{-1}), \tag{105}$$

where we have used the identity $u(G_1) = u(T_1 T_2) = T_1^{-1} T_2^{-1} = T_2^{-1} T_1^{-1} = G_1^{-1}$. Since in the adopted basis all the generators $G_1$, $G_2$ and $G_3$ are represented by unitary and symmetric matrices, we can choose the canonical CP, $X_{\mathbf{r}} = \mathbb{1}_{\mathbf{r}}$. Moreover, the Clebsh-Gordan coefficients are all real in this basis, and we will work with modular forms $Y(\tau)$ for which eq. (73) holds. By choosing minimal kinetic terms, we conclude that the theory is CP invariant when all Lagrangian parameters are real.

We choose the same field content and the same weight and representation assignment as in Lepton model II of ref. [2]. The matter chiral multiplets consist of three $SU(2)_L$ singlets $E^c$, three doublets $L$, two Higgs doublets $H_{u,d}$, transforming as

$$\rho_{E^c} = \mathbf{2} \oplus \mathbf{1}, \quad \rho_L = \mathbf{3}', \quad \rho_{H_u} = \rho_{H_d} = \mathbf{1},$$
$$k_{H_u} = k_{H_d} = 0, \quad k_{E_D^c} = -3, \quad k_{E_3^c} = k_L = -1. \tag{106}$$

Neutrino masses are described by the Weinberg operator. The superpotential of the lepton sector includes:

$$w_e = \alpha (E_D^c L Y_{\mathbf{3}'a}^{(4)})_{\mathbf{1}} H_d + \beta (E_D^c L Y_{\mathbf{3}'b}^{(4)})_{\mathbf{1}} H_d + \gamma (E_3^c L Y_{\mathbf{3}'})_{\mathbf{1}} H_d,$$
$$w_\nu = \frac{g_1}{\Lambda} (L L Y_{\mathbf{3}'})_{\mathbf{1}} H_u H_u + \frac{g_2}{\Lambda} (L L Y_{\mathbf{1}})_{\mathbf{1}} H_u H_u. \tag{107}$$

The phases of parameters $\alpha, \gamma$ and $g_1$ are unphysical and can be always removed, even before enforcing CP invariance.

We impose CP invariance, requiring also $\beta$ and $g_2$ to be real. At this level, the predictions of the model depend on five Lagrangian parameters plus the complex values of $\tau_1$ and $\tau_3$. From the superpotential and from the Clebsh-Gordan coefficients of $S_4 \times Z_2$, the charged lepton and neutrino mass matrices read:

$$M_e = \begin{pmatrix} \alpha(\sqrt{2} Y_{\mathbf{3}'a,2}^{(4)} - 2 Y_{\mathbf{3}'a,3}^{(4)}) + \beta(\sqrt{2} Y_{\mathbf{3}'b,2}^{(4)} - 2 Y_{\mathbf{3}'b,3}^{(4)}) & \sqrt{2}\alpha Y_{\mathbf{3}'a,1}^{(4)} + \sqrt{2}\beta Y_{\mathbf{3}'b,1}^{(4)} & -2\alpha Y_{\mathbf{3}'a,1}^{(4)} - 2\beta Y_{\mathbf{3}'b,1}^{(4)} \\ -\sqrt{2}\alpha Y_{\mathbf{3}'a,1}^{(4)} - \sqrt{2}\beta Y_{\mathbf{3}'b,1}^{(4)} & \alpha(\sqrt{2} Y_{\mathbf{3}'a,2}^{(4)} + 2 Y_{\mathbf{3}'a,3}^{(4)}) + \beta(\sqrt{2} Y_{\mathbf{3}'b,2}^{(4)} + 2 Y_{\mathbf{3}'b,3}^{(4)}) & 2\alpha Y_{\mathbf{3}'a,2}^{(4)} + 2\beta Y_{\mathbf{3}'b,2}^{(4)} \\ \gamma Y_1 & \gamma Y_2 & \gamma Y_3 \end{pmatrix} v_d,$$

$$M_\nu = \begin{pmatrix} g_1(\sqrt{2} Y_2 + Y_3) + g_2 Y_4 & \sqrt{2} g_1 Y_1 & g_1 Y_1 \\ \sqrt{2} g_1 Y_1 & g_1(-\sqrt{2} Y_2 + Y_3) + g_2 Y_4 & g_1 Y_2 \\ g_1 Y_1 & g_1 Y_2 & -2 g_1 Y_3 + g_2 Y_4 \end{pmatrix} \frac{v_u^2}{\Lambda}. \tag{108}$$

Here $Y_{\mathbf{3}'} = (Y_1, Y_2, Y_3)$ and $Y_{\mathbf{1}} = Y_4$ are weight 2 modular forms, while $Y_{\mathbf{3}'a}^{(4)} = (Y_{\mathbf{3}'a,1}^{(4)}, Y_{\mathbf{3}'a,2}^{(4)}, Y_{\mathbf{3}'a,3}^{(4)})$ and $Y_{\mathbf{3}'b}^{(4)} = (Y_{\mathbf{3}'b,1}^{(4)}, Y_{\mathbf{3}'b,2}^{(4)}, Y_{\mathbf{3}'b,3}^{(4)})$ are weight 4 modular forms. Their explicit expressions in terms of the second order theta constants are given in appendix B. We find a good agreement between the model predictions and the experimental data, for the following choice of parameters:

$$\tau_1 = -0.03376 + 1.11329i, \quad \tau_3 = -0.02376 + 0.50670i,$$
$$\beta/\alpha = -0.83991, \quad \gamma/\alpha = 0.01176, \quad g_2/g_1 = 1.58030,$$
$$\alpha v_d = 39.87894 \text{ MeV}, \quad g_1^2 v_u^2 / \Lambda = 6.07771 \text{ meV}. \tag{109}$$

---

[11]With an abuse of language we keep denoting by $N(H)$ also the projection of $N(H)$ at level two.

Table 2: The best fit values and the $1\sigma$ ranges of the charged lepton mass ratios and the lepton mixing parameters. The charged lepton mass ratios averaged over $\tan\beta$ [5] are taken from ref. [82], and we adopt the values of the lepton mixing parameters from NuFIT v5.0 with Super-Kamiokanda atmospheric data for normal ordering [83].

| Parameters | Best fit value and $1\sigma$ error |
|:---:|:---:|
| $m_e/m_\mu$ | $0.0048 \pm 0.0002$ |
| $m_\mu/m_\tau$ | $0.0565 \pm 0.0045$ |
| $\Delta m_{21}^2/10^{-5}\mathrm{eV}^2$ | $7.42^{+0.21}_{-0.20}$ |
| $\Delta m_{31}^2/10^{-3}\mathrm{eV}^2$ | $2.517^{+0.026}_{-0.028}$ |
| $\delta_{CP}/\pi$ | $1.0944^{+0.1500}_{-0.1333}$ |
| $\sin^2\theta_{12}$ | $0.304^{+0.012}_{-0.012}$ |
| $\sin^2\theta_{13}$ | $0.02219^{+0.00062}_{-0.00063}$ |
| $\sin^2\theta_{23}$ | $0.573^{+0.016}_{-0.020}$ |

The experimental $1\sigma$ ranges shown in table 2, with the exclusion of that referring to the Dirac CP phase $\delta_{CP}/\pi$, are the input data in our fit. Accordingly, the lepton masses and mixing parameters are determined to be:

$$\sin^2\theta_{12} = 0.3040\,, \quad \sin^2\theta_{13} = 0.02217\,, \quad \sin^2\theta_{23} = 0.5428\,, \quad \delta_{CP} = 1.57\pi\,,$$
$$\alpha_{21} = 0.19\pi\,, \ \ \alpha_{31} = 1.13\pi\,, \ \ m_e/m_\mu = 0.00480\,, \ \ m_\mu/m_\tau = 0.05735\,,$$
$$m_1 = 9.22 \text{ meV}\,, \quad m_2 = 12.62 \text{ meV}\,, \quad m_3 = 52.07 \text{ meV}\,,$$
$$m_\beta = 12.77 \text{ meV}\,, \quad m_{\beta\beta} = 10.74 \text{ meV}\,, \tag{110}$$

$m_\beta$ and $m_{\beta\beta}$ being the effective neutrino masses in beta decay and neutrinoless double beta decay, respectively. Note that the model predicts a Dirac CP phase $\delta_{CP}$ close to $3\pi/2$. The neutrino masses are of normal hierarchy type and they are quite tiny. All the experimental bounds from neutrino oscillations [83], tritium beta decays [84], neutrinoless double decay [85] and cosmology [86] are satisfied. We can make a further step and restrict the complex moduli to the one-dimensional invariant subspace $\tau_3 = \tau_1/2 = \tau_2/2$ contained in the region $\tau_1 = \tau_2$. We find that the observed lepton masses and mixing angles can still be accommodated [12]. The best fit values of the remaining input parameters are given by:

$$\tau_1 = -0.02827 + 1.17613i\,, \quad (\tau_1 = \tau_2 = 2\tau_3)$$
$$\beta/\alpha = -1.02608\,, \quad \gamma/\alpha = 0.01695\,, \quad g_2/g_1 = 1.42981\,,$$
$$\alpha v_d = 38.52395 \text{ MeV}\,, \quad g_1^2 v_u^2/\Lambda = 6.77168 \text{ meV}\,. \tag{111}$$

The masses and mixing parameters of leptons are predicted to be:

$$\sin^2\theta_{12} = 0.3036\,, \quad \sin^2\theta_{13} = 0.02215\,, \quad \sin^2\theta_{23} = 0.5291\,, \quad \delta_{CP} = 1.41\pi\,,$$
$$\alpha_{21} = 0.17\pi\,, \ \ \alpha_{31} = 1.13\pi\,, \ \ m_e/m_\mu = 0.00480\,, \ \ m_\mu/m_\tau = 0.05801\,,$$
$$m_1 = 10.08 \text{ meV}\,, \quad m_2 = 13.26 \text{ meV}\,, \quad m_3 = 51.26 \text{ meV}\,,$$
$$m_\beta = 13.40 \text{ meV}\,, \quad m_{\beta\beta} = 11.26 \text{ meV}\,. \tag{112}$$

We have comprehensively explored the parameter space of the model, within the invariant subspace $\tau_1 = \tau_2 = 2\tau_3$. Requiring the three lepton mixing angles and neutrino squared mass

---

[12]Notice that another possible one-dimensional subspace, $\tau_3 = 0$, can also fit the data well, except that $\sin^2\theta_{23}$ is slightly smaller than the $3\sigma$ lower bound of the experimental value.

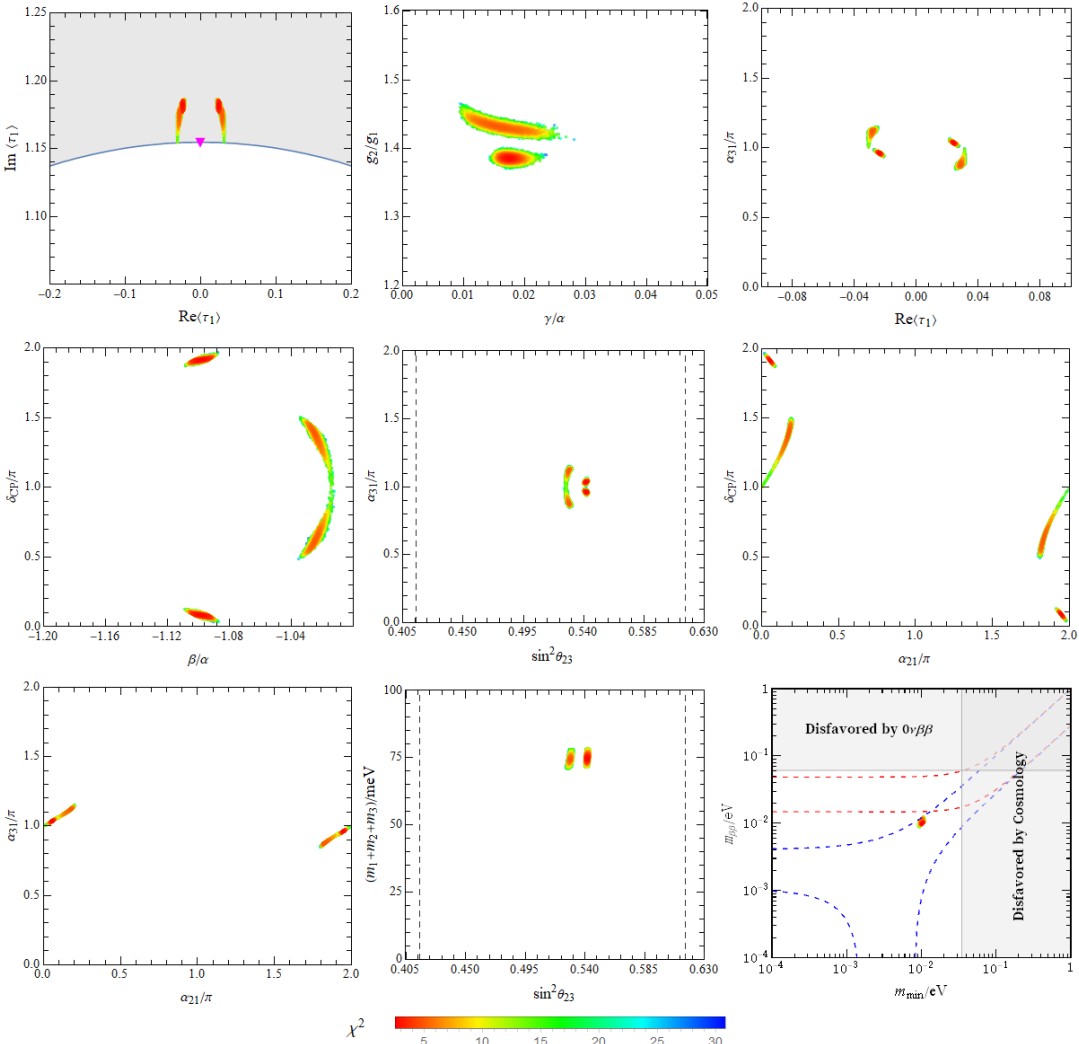

Figure 1: The correlations among the input free parameters, neutrino mixing angles and CP violating phases in the model discussed in the text, where the moduli $\tau$ are restricted to the subspace with $\tau_1 = \tau_2 = 2\tau_3$. The zero-dimensional fixed point $\tau = \frac{i}{\sqrt{3}} \left( \begin{smallmatrix} 2 & 1 \\ 1 & 2 \end{smallmatrix} \right)$ is marked by pink triangle in the $\tau_1$ plane .

splittings $\Delta m_{21}^2$, $\Delta m_{31}^2$ to lie in the experimentally allowed $3\sigma$ regions [83], we get the correlations between the free parameters and observable quantities shown in figure 1. It is worth noting that $\tau_1 = -0.02827 + 1.17613i$ is close to $2i/\sqrt{3} \approx 1.1547i$ and the VEVs of moduli that best reproduce the data are all clustered near the zero-dimensional fixed point $\frac{i}{\sqrt{3}} \left( \begin{smallmatrix} 2 & 1 \\ 1 & 2 \end{smallmatrix} \right)$. This point preserves CP and a $Z_2$ residual symmetry generated by $(\mathcal{ST})^2 \mathcal{TSV}$. We see that a small deviation from the CP-conserving point is sufficient to generate a sizable amount of CP violation.

In this model we can look numerically for the points preserving CP and compare them with those discussed analytically in the previous Section. By varying the value of $\tau$ in the modular subspaces with $\tau_1 = \tau_2, \tau_3 = 0$ or $\tau_1 = \tau_2 = 2\tau_3$, while keeping all other parameters fixed to their best fit values, we present the CP violating quantity $\Delta_{CP} \equiv \frac{1}{3}(|\sin \delta_{CP}| + |\sin \alpha_{21}| + |\sin \alpha_{31}|)$ as two heatmap plots in figure 2. The colors of the points in the figure change from blue to red, indicating that the $\Delta_{CP}$ is increasing: $\Delta_{CP} = 0$ corresponds to CP conservation and $\Delta_{CP} = 1$ corresponds to maximal CP violation. The CP conserved points (shown with dark blue colour)

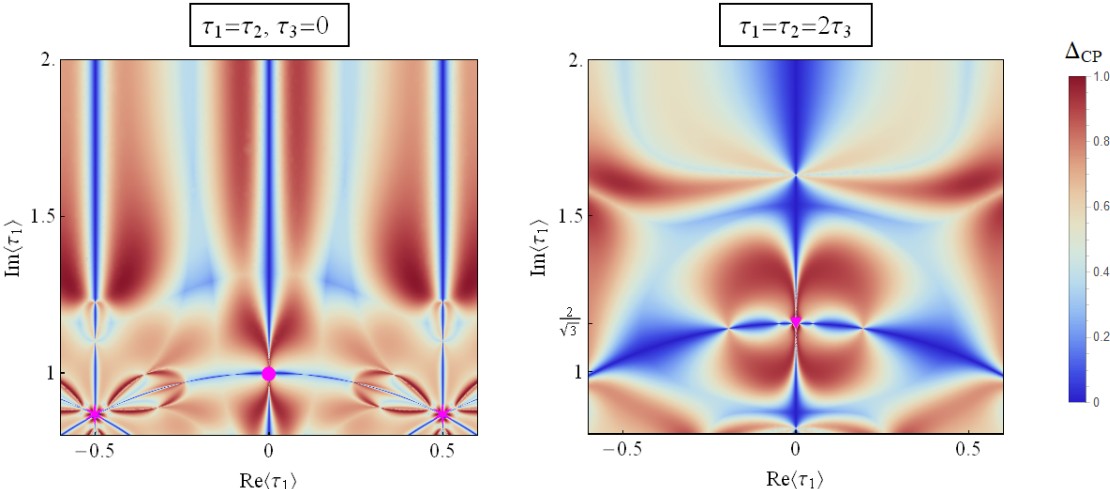

Figure 2: The distribution of CP violation measured by the average of the three CP phases $\Delta_{CP}$ in the subspace with $\tau_1 = \tau_2, \tau_3 = 0$ (left panel) and $\tau_1 = \tau_2 = 2\tau_3$ (respectively right panel). In the subspace with $\tau_3 = 0$, $|\text{Re}(\tau_1)| = 0, 1/2$ and $|\tau_1| = 1$ are CP-conserving points. In the subspace with $\tau_3 = \tau_1/2$, $|\text{Re}(\tau_1)| = 0$ and $|\tau_1| = 2/\sqrt{3}$ are CP-conserving points. The zero-dimensional fixed points $\begin{pmatrix} i & 0 \\ 0 & i \end{pmatrix}$, $\begin{pmatrix} \omega & 0 \\ 0 & \omega \end{pmatrix}$ and $\frac{i}{\sqrt{3}}\begin{pmatrix} 2 & 1 \\ 1 & 2 \end{pmatrix}$ are marked by pink dot, hexagonal star and triangle, respectively.

are indeed consistent with our analytic results. It is remarkable that significant CP violation can be induced for small deviation of $\tau$ from the zero-dimensional fixed points $\begin{pmatrix} i & 0 \\ 0 & i \end{pmatrix}$, $\begin{pmatrix} \omega & 0 \\ 0 & \omega \end{pmatrix}$ and $\frac{i}{\sqrt{3}}\begin{pmatrix} 2 & 1 \\ 1 & 2 \end{pmatrix}$.

# 7 Discussions

CP symmetry and its violation are key elements of a correct description of particle interactions. They are also crucial in explaining the baryonic asymmetry of the universe. CP violation has been observed in a rich variety of physical processes, but it is traceable to a unique source: a single observable phase in the CKM mixing matrix. A similar, yet-to-be-discovered, source can reside in the lepton mixing matrix, thus closely linking the asymmetry between the properties of particles and antiparticles to the features of the fermionic mass spectrum. CP transformations are a basic ingredient of any description of particle interactions. In theories invariant under the action of a local, continuous, gauge group, the existence of CP transformations, inverting the sign of commuting gauge charges is always guaranteed [87]. In general, up to topological terms, pure gauge interactions are automatically CP invariant, while Yukawa interactions are not. Nonetheless, the possibility that CP is a symmetry of the entire theory, including the Yukawa sector, is very appealing. The observed degree of CP violation would arise as a consequence of the choice of the vacuum and not from the adjustment of ad-hoc free parameters.

There is an interesting class of flavour models where the vacuum plays a key role in the description of fermion masses and mixing angles. Here both the flavour group and the SB sector have a common root. SB is parametrized by moduli, scalars taking values in a symmetric space of the type $G/K$. A discrete, modular subgroup $\Gamma$ of $G$, acting on $G/K$, plays the role of flavour symmetry. The symmetry associated with $\Gamma$ is a gauge symmetry and is related to the redundancy of the vacuum description. Physically inequivalent vacua are described by a fun-

damental domain $(G/K)/\Gamma$. Thus the vacuum is specified by a point in the multidimensional moduli space, up to a discrete modular transformation. To preserve the structure of $(G/K)/\Gamma$, CP transformations are to be searched among the nontrivial automorphisms of $\Gamma$. The existence of such automorphisms allows to enforce CP invariance and to provide a common origin of fermion masses, mixing angles and CP violating phases.

Pursuing a bottom-up approach, we have analyzed the allowed CP definitions in symplectic modular invariant theories, where $G = Sp(2g, \mathbb{R})$ and $K = U(g)$, starting from a complete classification of the automorphisms of the symplectic modular group $\Gamma = Sp(2g, \mathbb{Z})$. Notice that the symplectic modular group $Sp(2g, \mathbb{Z})$ coincides with $SL(2, \mathbb{Z})$ for the smallest genus $g = 1$. A unique possibility emerges when $g \geq 3$, while two are allowed for $g \leq 2$. We have also discussed the action of CP transformations on moduli, matter multiplets and modular forms, the building blocks for the construction of flavour models. In these theories, physically inequivalent vacua are described by a fundamental domain $\mathcal{F}_g$ in the Siegel upper half plane, whose explicit construction is known only at genus one and two. We have shown that in the interior of $\mathcal{F}_g$ CP is preserved only on the surface $\text{Re}(\tau) = 0$, while on its boundary there are infinite CP-conserving points. An interesting open problem is to establish whether all the points of the boundary are CP-conserving, like in genus one, or not.

Finally, we have shown how to combine all the previously discussed elements in the construction of a CP and symplectic invariant model of lepton masses at genus two. In the adopted framework, where the Kähler potential is minimal, the representations of the finite modular group are symmetric, the Clebsh-Gordan coefficients are all real and a suitable basis of modular forms is chosen, CP invariance is enforced by requiring that all Lagrangian parameters are real. In our model we manage to correctly reproduce the observed lepton masses and mixing angles by using five real free parameters. Neutrinos are Majorana particles, with a normally ordered mass spectrum. The model predicts all the three CP-violating phases, with the value of $\delta_{CP}$ approaching $3\pi/2$.

In our analysis, we have not attempt to determine dynamically the vacuum [88–90]. Rather, we have treated the moduli VEV as additional free parameters, optimized to maximize the agreement between data and predictions, with the hope of gaining some insight into the nature of the preferred vacuum. It is remarkable that the best values of moduli obtained in this way are very close to a point of enhanced symmetry, where both CP and some finite modular transformations are preserved. Thus, it suffices a small departure from a CP-conserving vacuum to generate sizable CP-violating effects. This confirms an intriguing behaviour already noticed in genus one constructions [91–93], where also the charged fermion hierarchy can benefit from the proximity to one such vacuum [94, 95]. If the previously outlined scenario is acceptable and provides a good enough description of the real world, we are confronted with a fascinating question: why is our universe living so close to a critical point?

# A The finite Siegel modular group $S_4 \times Z_2$

The finite modular group $S_4 \times Z_2$ can be generated by three generators: $G_1 \equiv T_1 T_2$, $G_2 \equiv T_3$, $G_3 \equiv S$ satisfying the multiplication rules:

$$G_1^2 = G_2^2 = G_3^2 = (G_1 G_2)^2 = (G_1 G_3)^3 = (G_1 G_2 G_3)^4 = 1. \tag{113}$$

The $S_4$ and $Z_2$ subgroups are generated by $\mathcal{S} = G_1$, $\mathcal{T} = (G_3 G_2)^4$ and $\mathcal{V} = (G_3 G_2)^3$ respectively, which obey the relations:

$$\mathcal{S}^2 = \mathcal{T}^3 = (\mathcal{S}\mathcal{T})^4 = 1, \quad \mathcal{V}^2 = 1, \quad \mathcal{S}\mathcal{V} = \mathcal{V}\mathcal{S}, \quad \mathcal{T}\mathcal{V} = \mathcal{V}\mathcal{T}. \tag{114}$$

The generators $G_{1,2,3}$ can be expressed in terms of $\mathcal{S}$, $\mathcal{T}$ and $\mathcal{V}$ as $G_1 = \mathcal{S}$, $G_2 = ((\mathcal{S}\mathcal{T})^2 \mathcal{T} \mathcal{S})^3 \mathcal{T} \mathcal{V}$, $G_3 = (\mathcal{S}\mathcal{T})^2 \mathcal{T} \mathcal{S}$.

The group $S_4 \times Z_2$ has four singlet representations $\mathbf{1}$, $\mathbf{1}'$, $\hat{\mathbf{1}}$, $\hat{\mathbf{1}}'$, two doublet representations $\mathbf{2}$, $\hat{\mathbf{2}}$, and four triplet representations $\mathbf{3}$, $\mathbf{3}'$, $\hat{\mathbf{3}}$ and $\hat{\mathbf{3}}'$. When constructing a CP and symplectic modular invariant model, it is more convenient to work in the basis of $X_{\mathbf{r}} = \mathbb{1}_{\mathbf{r}}$. Since the indicator $\mathrm{Ind}_{\mathbf{r}} = +1$ in all representations $\mathbf{r}$, such a basis can really be achieved. For the singlet representations, we have

$$
\begin{aligned}
\mathbf{1}\,(\hat{\mathbf{1}}): &\quad \mathcal{S} = \mathcal{T} = 1, \quad \mathcal{V} = 1\,(-1), \\
\mathbf{1}'\,(\hat{\mathbf{1}}'): &\quad \mathcal{S} = -1, \quad \mathcal{T} = 1, \quad \mathcal{V} = 1\,(-1).
\end{aligned}
\tag{115}
$$

In the doublet representations, the generators are represented by

$$
\mathbf{2}\,(\hat{\mathbf{2}}): \quad \mathcal{S} = \frac{1}{2}\begin{pmatrix} 1 & -\sqrt{3} \\ -\sqrt{3} & -1 \end{pmatrix}, \quad \mathcal{T} = \frac{1}{2}\begin{pmatrix} -1 & \sqrt{3} \\ -\sqrt{3} & -1 \end{pmatrix}, \quad \mathcal{V} = \mathbb{1}_2\,(-\mathbb{1}_2).
\tag{116}
$$

For the doublet representations, the generators are

$$
\mathbf{3}\,(\hat{\mathbf{3}}): \quad \mathcal{S} = \frac{1}{6}\begin{pmatrix} -3 & \sqrt{3} & 2\sqrt{6} \\ \sqrt{3} & -5 & 2\sqrt{2} \\ 2\sqrt{6} & 2\sqrt{2} & 2 \end{pmatrix}, \quad \mathcal{T} = \frac{1}{2}\begin{pmatrix} -1 & -\sqrt{3} & 0 \\ \sqrt{3} & -1 & 0 \\ 0 & 0 & 2 \end{pmatrix}, \quad \mathcal{V} = \mathbb{1}_3\,(-\mathbb{1}_3),
$$

$$
\mathbf{3}'\,(\hat{\mathbf{3}}'): \quad \mathcal{S} = -\frac{1}{6}\begin{pmatrix} -3 & \sqrt{3} & 2\sqrt{6} \\ \sqrt{3} & -5 & 2\sqrt{2} \\ 2\sqrt{6} & 2\sqrt{2} & 2 \end{pmatrix}, \quad \mathcal{T} = \frac{1}{2}\begin{pmatrix} -1 & -\sqrt{3} & 0 \\ \sqrt{3} & -1 & 0 \\ 0 & 0 & 2 \end{pmatrix}, \quad \mathcal{V} = \mathbb{1}_3\,(-\mathbb{1}_3).
\tag{117}
$$

It is easy to check that the representation matrices of $G_1$, $G_2$ and $G_2$ are unitary and symmetric in all irreducible representations. As a consequence, the CP symmetry for the automorphism $u_1$ is exactly the canonical CP in this basis with $X_{\mathbf{r}} = \mathbb{1}_{\mathbf{r}}$, as shown in Section 6.

The decomposition rules of the Kronecker product of two irreducible representations are necessary in model construction. We report the Kronecker products and the Clebsch-Gordan coefficients in the above CP basis in table 3. The notations $\alpha_i$ and $\beta_i$ refer to the elements of the first and the second representation of the product respectively.

## B    Siegel modular forms of genus $g = 2$ at level $n = 2$

There are five linearly independent Seigel modular forms $p_{0,1,2,3,4}$ at weight $k = 2$ and level $n = 2$, and they are form a quintet of the finite modular group $\Gamma_{2,2} \cong S_6$ [96]:

$$
\begin{aligned}
p_0 &= \Theta[00]^4(\tau) + \Theta[01]^4(\tau) + \Theta[10]^4(\tau) + \Theta[11]^4(\tau), \\
p_1 &= 2\left(\Theta[00]^2(\tau)\Theta[01]^2(\tau) + \Theta[10]^2(\tau)\Theta[11]^2(\tau)\right), \\
p_2 &= 2\left(\Theta[00]^2(\tau)\Theta[10]^2(\tau) + \Theta[01]^2(\tau)\Theta[11]^2(\tau)\right), \\
p_3 &= 2\left(\Theta[00]^2(\tau)\Theta[11]^2(\tau) + \Theta[01]^2(\tau)\Theta[10]^2(\tau)\right), \\
p_4 &= 4\Theta[00](\tau)\Theta[01](\tau)\Theta[10](\tau)\Theta[11](\tau),
\end{aligned}
\tag{118}
$$

where $\Theta$ is the second order theta constant defined by:

$$
\Theta[\sigma](\tau) = \sum_{m \in \mathbb{Z}^g} e^{2\pi i (m + \sigma/2)\tau(m + \sigma/2)^t},
\tag{119}
$$

where $\sigma = (\sigma_1, \sigma_2, \ldots, \sigma_g)$ are row vectors with $\sigma_i = 0, 1$. When we restrict $\tau$ to the two-dimensional modular subspace with $\tau_1 = \tau_2$, the relation $p_1(\tau) = p_2(\tau)$ is fulfilled, thus the modular forms space of weight 2 collapses into a four-dimensional subspace. The Siegel

Table 3: The Kronecker products and Clebsch-Gordan coefficients of the $S_4 \times Z_2$ group.

| $1 \otimes 2 = \hat{1} \otimes \hat{2} = 2, \quad 1 \otimes \hat{2} = \hat{1} \otimes 2 = \hat{2}$ | $1' \otimes 2 = \hat{1}' \otimes \hat{2} = 2, \quad 1' \otimes \hat{2} = \hat{1}' \otimes 2 = \hat{2}$ |
|---|---|
| $\mathbf{2}, \hat{\mathbf{2}} \sim \begin{pmatrix} \alpha\beta_1 \\ \alpha\beta_2 \end{pmatrix}$ | $\mathbf{2}, \hat{\mathbf{2}} \sim \begin{pmatrix} \alpha\beta_2 \\ -\alpha\beta_1 \end{pmatrix}$ |
| $1 \otimes 3 = 1' \otimes 3' = \hat{1} \otimes \hat{3} = \hat{1}' \otimes \hat{3}' = 3,$ <br> $1 \otimes \hat{3} = 1' \otimes \hat{3}' = \hat{1} \otimes 3 = \hat{1}' \otimes 3' = \hat{3}$ | $1 \otimes 3' = 1' \otimes 3 = \hat{1} \otimes \hat{3}' = \hat{1}' \otimes \hat{3} = 3',$ <br> $1 \otimes \hat{3}' = 1' \otimes \hat{3} = \hat{1} \otimes 3' = \hat{1}' \otimes 3 = \hat{3}'$ |
| $\mathbf{3}, \hat{\mathbf{3}} \sim \begin{pmatrix} \alpha\beta_1 \\ \alpha\beta_2 \\ \alpha\beta_3 \end{pmatrix}$ | $\mathbf{3}', \hat{\mathbf{3}}' \sim \begin{pmatrix} \alpha\beta_1 \\ \alpha\beta_2 \\ \alpha\beta_3 \end{pmatrix}$ |
| $2 \otimes 2 = \hat{2} \otimes \hat{2} = 1_s \oplus 1'_a \oplus 2_s, \quad 2 \otimes \hat{2} = \hat{1} \oplus \hat{1}' \oplus \hat{2}$ | |
| $\begin{aligned} \mathbf{1_s}, \hat{\mathbf{1}} &\sim \alpha_1\beta_1 + \alpha_2\beta_2 \\ \mathbf{1'_a}, \hat{\mathbf{1}}' &\sim \alpha_1\beta_2 - \alpha_2\beta_1 \\ \mathbf{2_s}, \hat{\mathbf{2}} &\sim \begin{pmatrix} \alpha_1\beta_2 + \alpha_2\beta_1 \\ \alpha_1\beta_1 - \alpha_2\beta_2 \end{pmatrix} \end{aligned}$ | |

| $2 \otimes 3 = \hat{2} \otimes \hat{3} = 3 \oplus 3', \quad 2 \otimes \hat{3} = \hat{2} \otimes 3 = \hat{3} \oplus \hat{3}'$ | $2 \otimes 3' = \hat{2} \otimes \hat{3}' = 3 \oplus 3', \quad 2 \otimes \hat{3}' = \hat{2} \otimes 3' = \hat{3} \oplus \hat{3}'$ |
|---|---|
| $\mathbf{3}, \hat{\mathbf{3}} \sim \begin{pmatrix} \sqrt{2}\alpha_1\beta_2 - \sqrt{2}\alpha_2\beta_1 - 2\alpha_1\beta_3 \\ \sqrt{2}\alpha_1\beta_1 + \sqrt{2}\alpha_2\beta_2 + 2\alpha_2\beta_3 \\ 2\alpha_2\beta_2 - 2\alpha_1\beta_1 \end{pmatrix}$ <br> $\mathbf{3}', \hat{\mathbf{3}}' \sim \begin{pmatrix} -\sqrt{2}\alpha_1\beta_1 - \sqrt{2}\alpha_2\beta_2 + 2\alpha_2\beta_3 \\ -\sqrt{2}\alpha_2\beta_1 + \sqrt{2}\alpha_1\beta_2 + 2\alpha_1\beta_3 \\ 2\alpha_2\beta_1 + 2\alpha_1\beta_2 \end{pmatrix}$ | $\mathbf{3}, \hat{\mathbf{3}} \sim \begin{pmatrix} -\sqrt{2}\alpha_1\beta_1 - \sqrt{2}\alpha_2\beta_2 + 2\alpha_2\beta_3 \\ -\sqrt{2}\alpha_2\beta_1 + \sqrt{2}\alpha_1\beta_2 + 2\alpha_1\beta_3 \\ 2\alpha_2\beta_1 + 2\alpha_1\beta_2 \end{pmatrix}$ <br> $\mathbf{3}', \hat{\mathbf{3}}' \sim \begin{pmatrix} \sqrt{2}\alpha_1\beta_2 - \sqrt{2}\alpha_2\beta_1 - 2\alpha_1\beta_3 \\ \sqrt{2}\alpha_1\beta_1 + \sqrt{2}\alpha_2\beta_2 + 2\alpha_2\beta_3 \\ 2\alpha_2\beta_2 - 2\alpha_1\beta_1 \end{pmatrix}$ |
| $3 \otimes 3 = 3' \otimes 3' = \hat{3} \otimes \hat{3} = \hat{3}' \otimes \hat{3}' = 1 \oplus 2 \oplus 3 \oplus 3',$ <br> $3 \otimes \hat{3} = 3' \otimes \hat{3}' = \hat{1} \oplus \hat{2} \oplus \hat{3} \oplus \hat{3}'$ | $3 \otimes 3' = \hat{3} \otimes \hat{3}' = 1' \oplus 2 \oplus 3 \oplus 3',$ <br> $3 \otimes \hat{3}' = 3' \otimes \hat{3} = \hat{1}' \oplus \hat{2} \oplus \hat{3} \oplus \hat{3}'$ |
| $\begin{aligned} \mathbf{1}, \hat{\mathbf{1}} &\sim \alpha_1\beta_1 + \alpha_2\beta_2 + \alpha_3\beta_3 \\ \mathbf{2}, \hat{\mathbf{2}} &\sim \begin{pmatrix} \sqrt{2}\alpha_2\beta_1 + \sqrt{2}\alpha_1\beta_2 - 2\alpha_1\beta_3 - 2\alpha_3\beta_1 \\ \sqrt{2}\alpha_2\beta_2 - \sqrt{2}\alpha_1\beta_1 + 2\alpha_2\beta_3 + 2\alpha_3\beta_2 \end{pmatrix} \\ \mathbf{3}, \hat{\mathbf{3}} &\sim \begin{pmatrix} \alpha_3\beta_2 - \alpha_2\beta_3 \\ \alpha_1\beta_3 - \alpha_3\beta_1 \\ \alpha_2\beta_1 - \alpha_1\beta_2 \end{pmatrix} \\ \mathbf{3}', \hat{\mathbf{3}}' &\sim \begin{pmatrix} \sqrt{2}\alpha_1\beta_2 + \sqrt{2}\alpha_2\beta_1 + \alpha_1\beta_3 + \alpha_3\beta_1 \\ \sqrt{2}\alpha_1\beta_1 - \sqrt{2}\alpha_2\beta_2 + \alpha_2\beta_3 + \alpha_3\beta_2 \\ \alpha_1\beta_1 + \alpha_2\beta_2 - 2\alpha_3\beta_3 \end{pmatrix} \end{aligned}$ | $\begin{aligned} \mathbf{1}', \hat{\mathbf{1}}' &\sim \alpha_1\beta_1 + \alpha_2\beta_2 + \alpha_3\beta_3 \\ \mathbf{2}, \hat{\mathbf{2}} &\sim \begin{pmatrix} \sqrt{2}\alpha_2\beta_2 - \sqrt{2}\alpha_1\beta_1 + 2\alpha_2\beta_3 + 2\alpha_3\beta_2 \\ -\sqrt{2}\alpha_2\beta_1 - \sqrt{2}\alpha_1\beta_2 + 2\alpha_1\beta_3 + 2\alpha_3\beta_1 \end{pmatrix} \\ \mathbf{3}, \hat{\mathbf{3}} &\sim \begin{pmatrix} \sqrt{2}\alpha_1\beta_2 + \sqrt{2}\alpha_2\beta_1 + \alpha_1\beta_3 + \alpha_3\beta_1 \\ \sqrt{2}\alpha_1\beta_1 - \sqrt{2}\alpha_2\beta_2 + \alpha_2\beta_3 + \alpha_3\beta_2 \\ \alpha_1\beta_1 + \alpha_2\beta_2 - 2\alpha_3\beta_3 \end{pmatrix} \\ \mathbf{3}', \hat{\mathbf{3}}' &\sim \begin{pmatrix} \alpha_3\beta_2 - \alpha_2\beta_3 \\ \alpha_1\beta_3 - \alpha_3\beta_1 \\ \alpha_2\beta_1 - \alpha_1\beta_2 \end{pmatrix} \end{aligned}$ |

modular forms of weight 2 and level 2 can be arranged into a singlet and a triplet of the finite Siegel modular subgroup $S_4 \times Z_2$:

$$\mathbf{3}': \quad Y_{\mathbf{3}'}(\tau) = \begin{pmatrix} \sqrt{3}\left(p_0(\tau) - 2p_1(\tau) - p_3(\tau) - 2p_4(\tau)\right) \\ p_0(\tau) - 2p_1(\tau) - p_3(\tau) + 6p_4(\tau) \\ \sqrt{2}\left(-p_0(\tau) - 4p_1(\tau) + p_3(\tau)\right) \end{pmatrix} \equiv \begin{pmatrix} Y_1(\tau) \\ Y_2(\tau) \\ Y_3(\tau) \end{pmatrix},$$

$$\mathbf{1}: \quad Y_{\mathbf{1}}(\tau) = p_0(\tau) + 3p_3(\tau) \equiv Y_4(\tau). \tag{120}$$

The weight four Siegel modular forms can be constructed from the tensor product of $Y_{\mathbf{1}}(\tau)$ and $Y_{\mathbf{3}'}(\tau)$. Using the Clebsch-Gordan coefficients listed in table 3, we find

$$
\begin{aligned}
\mathbf{1} : \quad & \begin{cases} Y_{\mathbf{1}a}^{(4)} = Y_{\mathbf{1}}Y_{\mathbf{1}} = Y_4^2 \,, \\ Y_{\mathbf{1}b}^{(4)} = (Y_{\mathbf{3}'}Y_{\mathbf{3}'})_{\mathbf{1}} = Y_1^2 + Y_2^2 + Y_3^2 \,, \end{cases} \\
\mathbf{2} : \quad & Y_{\mathbf{2}}^{(4)} = (Y_{\mathbf{3}'}Y_{\mathbf{3}'})_{\mathbf{2}} = \begin{pmatrix} 2\sqrt{2}Y_1Y_2 - 4Y_1Y_3 \\ \sqrt{2}(Y_2^2 - Y_1^2) + 4Y_2Y_3 \end{pmatrix} \,, \\
\mathbf{3} : \quad & Y_{\mathbf{3}}^{(4)} = (Y_{\mathbf{3}'}Y_{\mathbf{3}'})_{\mathbf{3}} = (0,0,0)^T \,, \\
\mathbf{3}' : \quad & \begin{cases} Y_{\mathbf{3}'a}^{(4)} = Y_{\mathbf{1}}Y_{\mathbf{3}'} = Y_4(Y_1, Y_2, Y_3)^T \,, \\ Y_{\mathbf{3}'b}^{(4)} = (Y_{\mathbf{3}'}Y_{\mathbf{3}'})_{\mathbf{3}'} = \begin{pmatrix} 2\sqrt{2}Y_1Y_2 + 2Y_1Y_3 \\ \sqrt{2}(Y_1^2 - Y_2^2) + 2Y_2Y_3 \\ Y_1^2 + Y_2^2 - 2Y_3^2 \end{pmatrix} \,, \end{cases}
\end{aligned}
\tag{121}
$$

where $Y_{\mathbf{3}'a}^{(4)}$ and $Y_{\mathbf{3}'b}^{(4)}$ denote the two independent weight 4 modular forms in the representation $\mathbf{3}'$.

# Acknowledgements

This project has received support in part by the European Union's Horizon 2020 research and innovation programme under the Marie Sklodowska-Curie grant agreement N° 674896 and 690575 and by the National Natural Science Foundation of China under Grant Nos 11975224, 11835013, 11947301, 12047502. The research of F. F. was supported in part by the INFN.

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
