# Peer review of "CP Symmetry and Symplectic Modular Invariance"

_SciPost Physics, doi:SciPost Phys. 10, 133 (2021)_

## Round 1 · Referee Report · Anonymous (Referee 1) · 2021-4-17

Report
The Authors develop the formalism of CP invariance in SUSY theories of flavour
based on symplectic modular invariance. In these theories the modular
symmetry is described by the the Siegel modular group
$\Gamma_g = Sp(2g, \mathbb{Z})$, while its quotient
$\Gamma_{g,n}$ -- the finite Siegel modular group -- plays the role of a
flavour symmetry, where $g$ and $n$ are positive integers,
$g$ being the genus of $\Gamma_g$. For $g=1$,
$Sp(2, \mathbb{Z})$ coincides with the modular group
$\Gamma \equiv SL(2,\mathbb{Z})$
of $2\times2$ integer matrices with determinant equal to 1.
$SL(2,\mathbb{Z})$, the inhomogeneous modular group
$\overline{\Gamma} \equiv PSL(2, \mathbb{Z})
\equiv SL(2,\mathbb{Z})/\{I,-I\}$, $I$ being the identity element,
and their respective quotient finite (modular) groups
$\Gamma_N$ and $\overline{\Gamma}_N$
($\Gamma_N$, e.g., is defined as the quotient
$\Gamma_N = SL(2,\mathbb{Z})/\Gamma(N)$,
$\Gamma(N)$ being the principal congruent (normal) subgroup of
$SL(2,\mathbb{Z})$ of level $N$), are widely being used
for flavour model building. For $N\leq 5$,
$\overline{\Gamma}_{2,3,4,5}$ are isomorphic
to the permutation groups $S_3$, $A_4$, $S_4$ and $A_5$,
while $\Gamma_{2,3,4,5}$ are isomorphic to their double covers
$S'_3 = S_3$, $A'_4 = T'$, $S'_4$ and $A'_5$. Thus,
$\Gamma_N$ and $\overline{\Gamma}_N$ play the role
of non-Abelian discrete flavour symmetry groups.
The modular invariance approach to the flavour problem,
put forward in 2017 and extensively investigated since the spring of 2018,
has opened up a new promising direction in the studies of this
highly challenging unresolved fundamental problem in particle
physics and the associated flavour model building.
In the approach, based on $SL(2,\mathbb{Z})$ and $\Gamma_N$
(or $PSL(2, \mathbb{Z})$ and $\overline{\Gamma}_N$), which have
been extensively studied, the Yukawa couplings and fermion mass matrices
in the Lagrangian of the theory are obtained from combinations
of modular forms which are holomorphic functions of the complex VEV of
a single complex scalar field -- the modulus $\tau$. The modular forms
together with the matter fields have specific transformation properties
under the action of the
modular symmetry group and furnish also irreducible representations
of the chosen finite modular (flavour symmetry)
group $\Gamma_N$ ($\overline{\Gamma}_N$). As a consequence of
the modular invariance, the fermion mass matrices
are expressed in terms of modular forms and certain number
of free, in general complex, constant parameters,
and have specific flavour structure.
In the minimal cases the number of free constants
is smaller than the number of the fitted observables. Thus, models of
flavour based on modular invariance have an increased predictive
power, constraining fermion masses, mixing and CP violating (CPV)
phases. Within the modular invariance approach, phenomenologically viable
models of lepton flavour with altogether 6 real and one
phase parameters including the complex VEV of $\tau$
(3 real constants for the 3 charged lepton masses and
3 real constants and one phase for the 3 neutrino masses, 3 lepton mixing
angles and 3 CP violating phases) have been constructed.
Viable quark flavour models have also been presented.
A very appealing feature of the modular invariance approach is that
in the simplest class of models, the VEV of the modulus $\tau$
is the only source of flavour symmetry breaking. Moreover,
it can also be the only source of breaking of CP symmetry. In this
case the constants present in the flavour models
are real, leading to significant reduction of the number of
free parameters.
The possibility of having the CP symmetry broken only by the VEV of
$\tau$ came to light after the formalism of combining CP and modular
symmetry was developed for flavour theories based on
$PSL(2, \mathbb{Z})$ and $SL(2, \mathbb{Z})$ modular symmetries.
In the paper under discussion the Authors extend this formalism
to the case of flavour theories based on the more sophisticated
symplectic modular symmetry described by the Siegel group
$\Gamma_g = Sp(2g, \mathbb{Z})$ with $g \geq 2$.
The extension is highly non-trivial and opens up
new possibilities for flavour model building.
In this case, in particular, the modulus $\tau$
and the spaces of the modular forms present in
the ``standard'' modular symmetry approach are replaced
by a $g\times g$ complex symmetric matrix of modulai and
larger Siegel modular form spaces, the dimensions of which depend
on the values of the genus $g$ and the level $n$.
The Authors identify the fundamental domain
in the space of moduli of the Siegel group. They
analyse the transformation properties of
the moduli, the Siegel modular forms and the matter fields
in the fundamental domain an its subspaces. They
formulate further the conditions of symplectic (Siegel)
modular invariance in SUSY theories of flavour.
The Authors perform next a very detailed comprehensive analysis
of how the CP summetry can be embedded in
these theories. They show, in particular, that in the case of
genus $g \geq 3$, there is a unique definition of the CP symmetry,
while for $g = 1,2$, two different definitions are possible.
They also show that a point $\tau$ in the interior
of the fundamental domain of the theory
is CP conserving if and only if it has a zero real part.
Investigating the consequences of the CP symmetry embedding
the Authors show that, as in the case of flavour theories
based on the ``standard'' modular symmetries
$PSL(2, \mathbb{Z})$ and $SL(2, \mathbb{Z})$,
the source of the CP symmetry breaking
can be just the VEVs of the moduli of the theory.
In this case the free constants present
in the Yukawa terms and correspondingly in the fermion mass
matrices are real. Finally, the Authors
construct a viable CP and symplectic invariant
model of lepton flavour based on
$Sp(2g, \mathbb{Z})$ with $g = 2$.
The model has five real constant parameters and
two moduli possessing complex VEVs which
break both the flavour and CP symmetries.
The data on the charged lepton masses
and neutrino oscillation parameters are reproduced by the model,
while the lightest neutrino mass, the type of neutrino mass spectrum,
leptonic CP violating phases and, correspondingly, the sum of
the neutrino masses and the effective Majorana mass in
neutrinoless double beta decay are predicted.
The paper contains original results of high scientific
quality, which are relevant for the physical problem
of flavour in particle physics. The analysis performed by the
Authors contains many subtle points and fine details I did not
comment on. The exposition is remarkably clear in spite
of the technical complexity of the subject.
I recommend the publication of the paper in SciPost.
based on symplectic modular invariance. In these theories the modular
symmetry is described by the the Siegel modular group
$\Gamma_g = Sp(2g, \mathbb{Z})$, while its quotient
$\Gamma_{g,n}$ -- the finite Siegel modular group -- plays the role of a
flavour symmetry, where $g$ and $n$ are positive integers,
$g$ being the genus of $\Gamma_g$. For $g=1$,
$Sp(2, \mathbb{Z})$ coincides with the modular group
$\Gamma \equiv SL(2,\mathbb{Z})$
of $2\times2$ integer matrices with determinant equal to 1.
$SL(2,\mathbb{Z})$, the inhomogeneous modular group
$\overline{\Gamma} \equiv PSL(2, \mathbb{Z})
\equiv SL(2,\mathbb{Z})/\{I,-I\}$, $I$ being the identity element,
and their respective quotient finite (modular) groups
$\Gamma_N$ and $\overline{\Gamma}_N$
($\Gamma_N$, e.g., is defined as the quotient
$\Gamma_N = SL(2,\mathbb{Z})/\Gamma(N)$,
$\Gamma(N)$ being the principal congruent (normal) subgroup of
$SL(2,\mathbb{Z})$ of level $N$), are widely being used
for flavour model building. For $N\leq 5$,
$\overline{\Gamma}_{2,3,4,5}$ are isomorphic
to the permutation groups $S_3$, $A_4$, $S_4$ and $A_5$,
while $\Gamma_{2,3,4,5}$ are isomorphic to their double covers
$S'_3 = S_3$, $A'_4 = T'$, $S'_4$ and $A'_5$. Thus,
$\Gamma_N$ and $\overline{\Gamma}_N$ play the role
of non-Abelian discrete flavour symmetry groups.
The modular invariance approach to the flavour problem,
put forward in 2017 and extensively investigated since the spring of 2018,
has opened up a new promising direction in the studies of this
highly challenging unresolved fundamental problem in particle
physics and the associated flavour model building.
In the approach, based on $SL(2,\mathbb{Z})$ and $\Gamma_N$
(or $PSL(2, \mathbb{Z})$ and $\overline{\Gamma}_N$), which have
been extensively studied, the Yukawa couplings and fermion mass matrices
in the Lagrangian of the theory are obtained from combinations
of modular forms which are holomorphic functions of the complex VEV of
a single complex scalar field -- the modulus $\tau$. The modular forms
together with the matter fields have specific transformation properties
under the action of the
modular symmetry group and furnish also irreducible representations
of the chosen finite modular (flavour symmetry)
group $\Gamma_N$ ($\overline{\Gamma}_N$). As a consequence of
the modular invariance, the fermion mass matrices
are expressed in terms of modular forms and certain number
of free, in general complex, constant parameters,
and have specific flavour structure.
In the minimal cases the number of free constants
is smaller than the number of the fitted observables. Thus, models of
flavour based on modular invariance have an increased predictive
power, constraining fermion masses, mixing and CP violating (CPV)
phases. Within the modular invariance approach, phenomenologically viable
models of lepton flavour with altogether 6 real and one
phase parameters including the complex VEV of $\tau$
(3 real constants for the 3 charged lepton masses and
3 real constants and one phase for the 3 neutrino masses, 3 lepton mixing
angles and 3 CP violating phases) have been constructed.
Viable quark flavour models have also been presented.
A very appealing feature of the modular invariance approach is that
in the simplest class of models, the VEV of the modulus $\tau$
is the only source of flavour symmetry breaking. Moreover,
it can also be the only source of breaking of CP symmetry. In this
case the constants present in the flavour models
are real, leading to significant reduction of the number of
free parameters.
The possibility of having the CP symmetry broken only by the VEV of
$\tau$ came to light after the formalism of combining CP and modular
symmetry was developed for flavour theories based on
$PSL(2, \mathbb{Z})$ and $SL(2, \mathbb{Z})$ modular symmetries.
In the paper under discussion the Authors extend this formalism
to the case of flavour theories based on the more sophisticated
symplectic modular symmetry described by the Siegel group
$\Gamma_g = Sp(2g, \mathbb{Z})$ with $g \geq 2$.
The extension is highly non-trivial and opens up
new possibilities for flavour model building.
In this case, in particular, the modulus $\tau$
and the spaces of the modular forms present in
the ``standard'' modular symmetry approach are replaced
by a $g\times g$ complex symmetric matrix of modulai and
larger Siegel modular form spaces, the dimensions of which depend
on the values of the genus $g$ and the level $n$.
The Authors identify the fundamental domain
in the space of moduli of the Siegel group. They
analyse the transformation properties of
the moduli, the Siegel modular forms and the matter fields
in the fundamental domain an its subspaces. They
formulate further the conditions of symplectic (Siegel)
modular invariance in SUSY theories of flavour.
The Authors perform next a very detailed comprehensive analysis
of how the CP summetry can be embedded in
these theories. They show, in particular, that in the case of
genus $g \geq 3$, there is a unique definition of the CP symmetry,
while for $g = 1,2$, two different definitions are possible.
They also show that a point $\tau$ in the interior
of the fundamental domain of the theory
is CP conserving if and only if it has a zero real part.
Investigating the consequences of the CP symmetry embedding
the Authors show that, as in the case of flavour theories
based on the ``standard'' modular symmetries
$PSL(2, \mathbb{Z})$ and $SL(2, \mathbb{Z})$,
the source of the CP symmetry breaking
can be just the VEVs of the moduli of the theory.
In this case the free constants present
in the Yukawa terms and correspondingly in the fermion mass
matrices are real. Finally, the Authors
construct a viable CP and symplectic invariant
model of lepton flavour based on
$Sp(2g, \mathbb{Z})$ with $g = 2$.
The model has five real constant parameters and
two moduli possessing complex VEVs which
break both the flavour and CP symmetries.
The data on the charged lepton masses
and neutrino oscillation parameters are reproduced by the model,
while the lightest neutrino mass, the type of neutrino mass spectrum,
leptonic CP violating phases and, correspondingly, the sum of
the neutrino masses and the effective Majorana mass in
neutrinoless double beta decay are predicted.
The paper contains original results of high scientific
quality, which are relevant for the physical problem
of flavour in particle physics. The analysis performed by the
Authors contains many subtle points and fine details I did not
comment on. The exposition is remarkably clear in spite
of the technical complexity of the subject.
I recommend the publication of the paper in SciPost.

---

## Round 1 · Referee Report · Anonymous (Referee 2) · 2021-5-13

Report
The paper "CP Symmetry and Symplectic Modular Invariance" by Ding, Feruglio and Liu discusses the extension of the Siegel modular group Sp$(2g,\mathbb{Z})$ of so-called genus $g$ by a CP symmetry. The Siegel modular approach has been suggested by the authors in a previous publication (arXiv:2010.07952) as the basic ingredient to explain masses and mixings especially in the lepton sector of the (supersymmetric extension of the) Standard Model of particle physics. In this paper, the authors extend their previous case by CP.
The paper first reviews the Siegel modular group $\Gamma_g=$ Sp$(2g,\mathbb{Z})$, its principal congruence subgroups $\Gamma_g(n)$ of level $n$, and the resulting finite quotient groups $\Gamma_{g,n}=\Gamma_g/\Gamma_g(n)$. These finite groups $\Gamma_{g,n}$, known as finite Siegel modular groups, and their subgroups are investigated as non-Abelian flavor symmetries.
To do so, a clear introduction into Siegel modular forms and modular invariant (supersymmetric) theories is given in section 2.
Then, CP is introduced, with a special focus on consistency conditions between CP and the Siegel modular group. It is shown that for $g=1,2$ there are two independent definitions of CP available, while for $g>2$ there exists a unique definition of CP. Having defined CP (as a $\mathbb{Z}_2$ outer automorphism of $\Gamma_g$), the transformations of the moduli, matter fields, and modular forms are given in all details. Furthermore, the conditions for a CP invaraint theory are spelled out.
In the next section, section 4, the authors analyze points in moduli space of unbroken CP and the implications on mass matrices (for neutrinos and charged leptons, for example).
Section 5 is devoted to the discussion of CP acting on subspaces in moduli space that are left invariant under certain subgroups of $\Gamma_g$ for $g=2$.
This is the starting point of model-building, as initiated in section 6, where a certain subspace in moduli space is chosen: by choosing the genus $g=2$, the level $n=2$ and the moduli $\tau_1=\tau_2$, the finite Siegel modular group $S_6$ breaks to $S_4 \times \mathbb{Z}_2$, which is used as the finite non-Abelian flavor symmetry of the proposed model. Certain additional assumptions have to be made (concerning for example the representations and the modular weights of leptons and Higgs doublets). This defines their model by the resulting superpotential as given in equation (6.3), where the right-handed neutrinos obtain masses via the Weinberg operator. From there, the mass matrices are computed. This yields the lepton masses and the PMNS matrix with its mixing angles and CP phases. It is shown that under the assumption of moduli stabilization at a very specific point in moduli space (close to a symmetry enhanced point), the model is in agreement with all experimental bounds.
The paper is very well written, contains many novel ideas that open up the avenue for further research in order to explain the flavor structure of the Standard Model. Furthermore, the approach based on Siegel modular symmetries is well-motivated as it is closely connected to various approaches in string theory (like magnetized branes and toroidal orbifolds). Hence, this paper is beneficial for both communities: bottom-up flavor model-buidling and string theory.
In conclusion, I highly recommend this paper for publication in SciPost.
The paper first reviews the Siegel modular group $\Gamma_g=$ Sp$(2g,\mathbb{Z})$, its principal congruence subgroups $\Gamma_g(n)$ of level $n$, and the resulting finite quotient groups $\Gamma_{g,n}=\Gamma_g/\Gamma_g(n)$. These finite groups $\Gamma_{g,n}$, known as finite Siegel modular groups, and their subgroups are investigated as non-Abelian flavor symmetries.
To do so, a clear introduction into Siegel modular forms and modular invariant (supersymmetric) theories is given in section 2.
Then, CP is introduced, with a special focus on consistency conditions between CP and the Siegel modular group. It is shown that for $g=1,2$ there are two independent definitions of CP available, while for $g>2$ there exists a unique definition of CP. Having defined CP (as a $\mathbb{Z}_2$ outer automorphism of $\Gamma_g$), the transformations of the moduli, matter fields, and modular forms are given in all details. Furthermore, the conditions for a CP invaraint theory are spelled out.
In the next section, section 4, the authors analyze points in moduli space of unbroken CP and the implications on mass matrices (for neutrinos and charged leptons, for example).
Section 5 is devoted to the discussion of CP acting on subspaces in moduli space that are left invariant under certain subgroups of $\Gamma_g$ for $g=2$.
This is the starting point of model-building, as initiated in section 6, where a certain subspace in moduli space is chosen: by choosing the genus $g=2$, the level $n=2$ and the moduli $\tau_1=\tau_2$, the finite Siegel modular group $S_6$ breaks to $S_4 \times \mathbb{Z}_2$, which is used as the finite non-Abelian flavor symmetry of the proposed model. Certain additional assumptions have to be made (concerning for example the representations and the modular weights of leptons and Higgs doublets). This defines their model by the resulting superpotential as given in equation (6.3), where the right-handed neutrinos obtain masses via the Weinberg operator. From there, the mass matrices are computed. This yields the lepton masses and the PMNS matrix with its mixing angles and CP phases. It is shown that under the assumption of moduli stabilization at a very specific point in moduli space (close to a symmetry enhanced point), the model is in agreement with all experimental bounds.
The paper is very well written, contains many novel ideas that open up the avenue for further research in order to explain the flavor structure of the Standard Model. Furthermore, the approach based on Siegel modular symmetries is well-motivated as it is closely connected to various approaches in string theory (like magnetized branes and toroidal orbifolds). Hence, this paper is beneficial for both communities: bottom-up flavor model-buidling and string theory.
In conclusion, I highly recommend this paper for publication in SciPost.

---

## Editorial Decision

published